# Utilization of Polymer-Lipid Hybrid Nanoparticles for Targeted Anti-Cancer Therapy

**DOI:** 10.3390/molecules25194377

**Published:** 2020-09-23

**Authors:** Ayeskanta Mohanty, Saji Uthaman, In-Kyu Park

**Affiliations:** 1Department of Biomedical Sciences, Chonnam National University Medical School, 264, Seoyang-ro, Jeollanam-do 58128, Korea; ayeskanta99@gmail.com; 2Department of Polymer Science and Engineering, Chungnam National University, 99 Daehak-ro, Yuseoung-gu, Daejeon 34134, Korea

**Keywords:** cancer, nanotechnology, polymeric nanoparticles, liposomes, polymer-lipid hybrid nanoparticles (PLHNPs), targeting ligands, anticancer therapy

## Abstract

Cancer represents one of the most dangerous diseases, with 1.8 million deaths worldwide. Despite remarkable advances in conventional therapies, these treatments are not effective to completely eradicate cancer. Nanotechnology offers potential cancer treatment based on formulations of several nanoparticles (NPs). Liposomes and polymeric nanoparticle are the most investigated and effective drug delivery systems (DDS) for cancer treatment. Liposomes represent potential DDS due to their distinct properties, including high-drug entrapment efficacy, biocompatibility, low cost, and scalability. However, their use is restricted by susceptibility to lipid peroxidation, instability, burst release of drugs, and the limited surface modification. Similarly, polymeric nanoparticles show several chemical modifications with polymers, good stability, and controlled release, but their drawbacks for biological applications include limited drug loading, polymer toxicity, and difficulties in scaling up. Therefore, polymeric nanoparticles and liposomes are combined to form polymer-lipid hybrid nanoparticles (PLHNPs), with the positive attributes of both components such as high biocompatibility and stability, improved drug payload, controlled drug release, longer circulation time, and superior in vivo efficacy. In this review, we have focused on the prominent strategies used to develop tumor targeting PLHNPs and discuss their advantages and unique properties contributing to an ideal DDS.

## 1. Introduction

Cancer is a dangerous illness and a public health concern. It can affect any organ in the body and is characterized by uncontrolled growth and division of abnormal cells. Successive gene mutations lead to dysregulation of many regulatory proteins, resulting in uncontrolled tumor growth [1]. However, the extracellular environment also plays a vital role in the development and progression of cancer [2]. Various strategies are used in cancer treatment, each with its own benefits and harms. Conventional therapeutic approaches that are extensively used clinically include surgery, chemotherapy, and radiotherapy [3,4]. These modalities are ineffective in the eradication of cancer completely. High-intensity radiation and surgical interventions can trigger tissue infections, inflammation, and DNA damage in healthy cells, which cannot be repaired, resulting in metastasis and recurrence [5]. Similarly, non-specific drug distribution and poor pharmacokinetics of chemotherapeutic drugs limit the therapeutic effectiveness and susceptibility to multi-drug resistance (MDR) during treatments [3]. Therefore, an effective cancer treatment requires a selective drug delivery system (DDS) to address the limitations of conventional therapies.

The potential of nanotechnology is based on the use of several engineered nanoparticles (NPs) for controlled drug delivery. The NPs are 1–1000 nanometers (nm) in diameter in size. The unique physiochemical properties of NPs, such as size, shape, surface charge, and surface area represent an advantage over current anti-cancer treatments. These properties have attracted the attention of researchers investigating cancer therapeutics and diagnostic applications. NPs increase the bioavailability of hydrophobic chemotherapeutics by improving their aqueous solubility, prolonging the circulation time within the body, and the site-specific delivery of therapeutic cargo by limiting the non-tumor-specific accumulation of the drug in healthy tissues. NPs carry a high payload and an effective DDS, representing a compelling drug delivery platform for the prevention [6], diagnosis [7,8], and treatment of cancer [9,10]. Many versatile NP-based systems, such as liposomes, micelles, polymers, inorganic NPs, and nanogels, are used as carriers for the delivery of drugs, genes, proteins, peptides, and imaging agents to the tumor site [11,12,13,14,15,16]. Polymeric and lipid-based NPs are the most effective DDSs due to their great success in systemic delivery of drug [17].

Liposomes are spherical fat bubbles with a lipid bilayer membrane consisting of both hydrophilic and lipophilic lipid molecules. The lipid membrane is composed of natural or synthetic phospholipids. Besides phospholipids, other lipids, such as cholesterol, are used to formulate NPs with increased fluidity, which results in increased biostability and permeability inside the body. Further, their circulation time can be improved by modifying the surface with polyethylene glycol (PEG) to form PEGylated or stealth liposomes. The remarkable advantages of liposomes are attributed to both the hydrophilic and hydrophobic drug encapsulation into the lipophilic and lipophobic compartments, which provides a flexible surface with multiple targeting ligands to actively target cancer cells [18]. The liposomes exhibit biocompatibility, biodegradability, ability for targeted drug delivery, and large-scale production for application in cancer research. Currently, 16 clinically approved liposomal drugs are available, for example, AmBiosome (amphotericin B), Visudyne (verteporfin), DepoCyt (cytarabine), DaunoXome (daunorubicin), Visudyne (verteporfin), and DepoDur (morphine) [19]. However, a few disadvantages limit the use of liposomes in biomedical applications. First, oxidation and hydrolysis of liposomal phospholipids lead to degradation of the liposomal structure, resulting in a burst release of the drug. Second, the lipid peroxidation changes the properties of the lipid membrane, such as rigidity and cellular permeability. The physical and chemical instability reduce the shelf life of the liposomes. Finally, accidental leakage, fusion of encapsulated drugs with lipids, reduced efficiency of hydrophobic drug encapsulation, storage instability, and high production cost hinder liposomal applications. However, polymer-based nanosystems stable inside the cells, and show controlled drug release behavior compared with liposomes [20,21,22]. The polymer-based NPs provide an easy and cost-effective formulation. The different chemical modifications with polymers diversify the various polymeric nano-carriers formed such as polymeric micelles, drug-polymer conjugates, and polymeric NPs. Biodegradable polymers, such as synthetic polymer poly-lactic acid (PCL), poly (lactic-co-glycolic acid) (PLGA), and the natural polymer chitosan are mostly used in cancer therapeutics [23,24]. However, the disadvantages of polymeric NPs include the need for organic solvents for synthesis, limited payload, and moderate circulation half-life [25]. To optimize the nanocarriers, scientists have combined the structural components of liposomes and polymeric NPs to develop hybrid carriers.

Polymer-lipid hybrid nanoparticles (PLHNPs) combine the advantages of both polymeric NPs and liposomes. This hybrid system overcomes the limitations of lipids and polymers and offers enormous potential in the field of nanomedicine [26]. Figure 1 represents different biomedical applications of PLHNPs. PLHNPs contain three major components: (1) a hydrophobic/hydrophilic polymeric core encapsulating both hydrophobic and hydrophilic drugs effectively and resulting in sustained release kinetics; (2) a lipid shell surrounding the polymeric core with high biocompatibility, overall stability, and reduced retention of the drug inside the polymeric core; (3) and an outer component, consisting of a lipid-polyethylene glycol (PEG), which is covered by a lipid layer to enhance steric stabilization, prolong circulation time, and prevent immune recognition [27]. The inner polymeric matrix and lipid layer are associated with van der Waals forces, electrostatic, hydrophobic, and noncovalent interactions, whereas the hydrophilic polymer core is generally conjugated with the surrounding lipid layers by covalent bonds [28,29,30]. Usually, two distinct synthetic approaches are available for PLHNPs, including a one-step and a two-step. In the one-step approach, either the free polymeric core or both the polymeric core and hydrophobic drug are dissolved in an appropriate solvent, and the prepared lipid shell and lipid-PEG are mixed in a single pot with the drug-loaded polymeric core and aqueous lipid solution via nanoprecipitation and self-assembly. In the two-step approach, the polymeric core and lipid shell are first prepared separately and then fused together to form PLHNPs [25,31]. Figure 2A,B represents one-step and two-step reactions in PLHNP synthesis, respectively. PLHNPs often show distinct benefits compared with those of traditional drug carriers, such as improved physiological properties, favorable release kinetics, superior capability to encapsulate both hydrophilic and hydrophobic therapeutic drugs, enormous plasma stability, easy formulation with a range of lipids and biodegradable polymers, differences in structural configuration, longer circulation times, cellular and molecular recognition, and enhanced biocompatibility and biodegradability [26,28,32]. The advantages of PLHNPs are shown in Figure 3.

## 2. Types of Polymer-Lipid Hybrid Nanoparticles

PLHNPs are classified depending on the differences in their lipid and polymer configurations. In each hybrid structure, the structural integrity of the polymers enhances the overall particle stability and improves the drug release kinetics. However, lipids often enhance drug loading and improve particle biocompatibility. Thus, the advanced hybrid NPs often exhibit higher and prolonged in vivo activity. The structural advantages of different PLHNP formulations are shown in Figure 4. The different hybrid structures represent effective delivery systems compared with either of their components alone. The different types of PLHNPs with structural arrangements are as follows.

### 2.1. Polymer Core Lipid-Shell Hybrid Nanoparticles (PCLHNPs)

The polymer core lipid-shell hybrid system (PCLHNP) consists of a partial polymer core, which is covered by a unilayer or bilayer lipid shell, as shown in Figure 4A. The polymeric core enhances the stability of the lipid layers. The biodegradable polymeric core and the stability of the lipids collectively provide an effective delivery system for treatment of a variety of cancers and are one of the easiest forms among all PLHNPs [33,34]. Because the amphiphilicity of polymers and lipids facilitates entrapment of both hydrophilic and hydrophobic drugs inside the hybrid system, this type of hybrid system is mainly synthesized via the emulsion method. For example, multi-layered PCLHNPs were used to deliver three types of chemotherapeutic drugs: oxaliplatin, camptothecin, and 5-fluorouracil (5-FU). The inner core of PCLHNP contains polylactic acid (PCL) and methoxy polyethylene glycol (mPEG-diblock copolymer), which encapsulates the hydrophobic camptothecin and hydrophilic 5-FU. Further, the inner core is covered by three types of lipid layers, lecithin, cholesterol, and DSPE-PEG (1,2-distearoyl-snglycero-3-phosphoethanilamine-*N*-poly (ethylene glycol)), which carry oxaliplatin through their lipid-polymer interface area [35]. Such a hybrid system was first formulated and applied in biotechnological and biomedical applications, such as biosensors and immunology kits [33,36]. Such NPs have been used recently for the co-delivery of a drug resistance inhibitor and an anticancer drug. Combining the delivery of these two entities inhibits the development of anti-cancer drug resistance [37].

PCLHNPs can be used to control multiple factors related to drug loading and encapsulation efficiency via synthetic methods, drug solubility, drug miscibility in the polymer and lipid phases, charge interaction between lipids and drugs, and the polymer: lipid ratio.

### 2.2. Monolithic Lipid-Polymer Hybrid Nanoparticles (MLPHNPs)

Mixed lipid-polymer hybrid NPs are also called monolithic lipid-polymer hybrid NPs (MLPHNPs). This hybrid system is formulated by mixing lipids throughout the polymeric core matrix, as shown in Figure 4C [27]. This nanoparticle system acts as a colloidal carrier for drug delivery [38]. Phospholipids are used as an integral component of this hybrid structure, which forms the carrier-like structure. However, the PEG chain modification with the lipid layer is attributed to the non-flexible nature of phospholipid. The mixing ratios of the polymer and lipid can be adjusted to reduce the systemic toxicity inside the body. For example, a negatively charged monolithic hybrid system mixed with rhamnolipid and phospholipid loaded with 10.2% of amoxicillin and an adhesion material pectin sulfate showed encapsulation efficiency of 93% with complete drug release in 24 h based on selective lipid mixing of rhamnolipid and phospholipid [39].

### 2.3. Core-Shelltype Hollow Lipid-Polymer Lipid Hybrid Nanoparticle (CSLPLHNPs)

Core-shell type hollow lipid-polymer lipid hybrid nanoparticles (CSLPLHNPs) consist of a hollow inner core comprising positively charged inner lipid layers, followed by a middle hydrophobic polymer layer (mostly PLGA layer), and an outer PEG lipid layer (neutral charge) covered by the polymer matrix, as shown in Figure 4B. They exhibit the properties of both polymer lipoplexes and PLGA NPs [26]. Mainly, these particles are synthesized via emulsification solvent evaporation. The hollow space between the lipids and the polymer is filled with water or aqueous buffer. Compared to the polymeric core, the innermost cationic zone of lipids entraps anionic drugs more efficiently [40]. The outer PEG lipid layer enhances the in vivo stability and protects the inner polymer core from microphase reorganization [27]. Occasionally, a combination of lipids is used to form the outermost layer of these hybrid NPs to stabilize the particle formulation. One of the lipids might self-assemble with the hydrophilic head facing the aqueous phase; however, its tail faces the polymeric core, and the remainder forms a complex lipid-PEG structure [27]. During the formation of such a complex structure, the concentration of lipids affects the formation of micelles and liposomes. The low concentration of lipids prevents such particle formation. The choice of the inner cationic lipids, the outer density of the PEG chain, and the molecular weight of the polymer are crucial for the formation of CSLPLHNPs in these hybrid structures [27]. CSLPLHNPs are constructed via a double-emulsion solvent evaporation. Shi. et al. used the self-assembly technique to entrap siRNAs in CSLPLHNPs. This hybrid system contains four layers, a positively charged hollow lipid layer, a hydrophobic PLGA polymeric middle layer, an outer neutral lipid layer (which forms an interface with the polymer PLGA), and an outer PEG layer. In the formulation, the inner cationic lipids entrapped the siRNA more efficiently than the polymeric matrix [41]. This type of PLHNP delivers two drugs at a time. The structural arrangement of CSHLPLHNPs facilitates the co-delivery of an siRNA and a small drug to overcome multi-drug resistance [42].

### 2.4. Polymer-Caged Liposome Hybrid Nanoparticles (PCLNPs)

The structural arrangement of polymer-caged liposomes involves polymer coating on the surface of the liposomes, as shown in Figure 4D. Besides, surface modification with different polymers enhanced the functionality of this nanosystem. Among all hybrid formulations, polymer-caged liposomes often exhibit the highest stability and stimuli-responsive drug release. Lee et al. modified the liposome surface using cholesterol functionalized polyacrylic acid (PAA) polymer, in which the carboxylic group of PAA is crosslinked to and forms an amide bond with 2,2-(ethylenedioxy) bis (ethylamine), a pH-responsive linker [43]. In this type of hybrid system, the polymer cage protects from the outer environment, and a stimulus triggers sustained drug release. The polymer coating confers various benefits, such as improved liposomal stability, enhanced drug release (at lower pH value), and improved drug-entrapment efficiency.

### 2.5. Cell Membrane-Camouflaged PLHNPs (CCPLHNPs)

The cell membranes are coated over polymer-lipid hybrid NPs to form membrane-camouflaged PLHNPs, as shown in Figure 4E. They are also known as biomimetic NPs owing to their surface chemistry, which mimics the natural membranes [44]. Cell membranes are coated over NPs mostly by extrusion method. Red blood cells (RBC) coated with NPs are natural vehicles for drug delivery and can easily avoid macrophage uptake [45]. Drugs are loaded into the hydrophobic polymeric matrix; however, the lipid density in the membrane increases the sustained drug release compared with both liposomes and polymeric NPs. This type of hybrid nanosystem easily overcomes the biological barrier and efficiently acts as a long circulating vehicle with higher in vivo stability compared with that of PEG alone because of membrane coating [46,47]. Even though this hybrid system has many advantages over simple polymer-lipid hybrid NPs, it is limited by the presence of different surface antigens on patients’ RBCs with different blood groups. The erythrocyte membrane camouflaged PLHNPs may thus crossmatch at the time of blood transfusion [45].

## 3. Targeting

For the design of an effective nanocarrier-based DDS, two essential factors should be considered. First, after administration, drugs should be able to reach the required site with minimal loss of activity in blood circulation. Second, drugs should specifically destroy cancer cells without affecting healthy cells [48]. Many strategies, including passive and active targeting, have thus been utilized to enable the efficient delivery of drugs to cancer cells. [49,50,51]. These strategies are used for the development of targeted anti-cancer therapies. These therapies target cancer-specific genes, proteins, and tumor microenvironment (TME). Here, we discuss the PLHNP-based passive and active targeting strategies.

### Passive Targeting

Passive targeting exploits the unique pathophysiology of tumor vessels, which facilitates NP accumulation in the tumor tissue. Tumor vessels are highly unstable, which results in the interstitial gap between the endothelial cells and the lymphatic system. The leaky vasculature of blood vessels is related to enhanced permeability and retention (EPR) effect [52]. The NP size and surface are crucial parameters that impact the EPR effect. It is reported that particles up to 100 nm in diameter migrate into the tumor region [52]. Recently, cisplatin-loaded lipid-chitosan hybrid NPs measuring 200 nm in size were targeted via EPR effect leading to prolonged circulation [50]. Khan et al. developed formulations with different lipid-to-chitosan ratio to optimize the particle size. Among the developed formulations, small NPs, measuring 181 nm in size and 20.5 mV in zeta-potential, resulted in a spherical lipoplex structure with 89.2% cisplatin loading efficiency [50]. The small-sized particles provide enhanced mean residence time and a longer half-life. Another study explained that the drug-loaded ultra-small polymer-lipid hybrid NPs, measuring less than 25 nm in size, often show deep tissue penetration and enhanced stability [53]. Besides size, the surface charge of NPs plays an important role in their in vivo properties. Mostly, negatively charged particles exhibit a strong interaction with cell membranes, and thereby an enhanced intercellular uptake. The lipid-PEG outer layer of PLHNPs enhances their circulation time and in vivo stability by allowing interaction with cells. This strategy was applied to create paclitaxel- and etoposide-loaded hybrid NPs for the treatment of osteosarcoma. The particles were spherical with a diameter of 190 nm and a surface charge of −22.8 mV, which indicate good physical ability. The particles showed increased cellular internalization and enhanced, prolonged blood circulation in tumor tissue [54]. A similar study developed PLHNPs to deliver lysozyme and evaluated their in vivo stability. The lysozyme loaded into the polylactic acid (PCL) was further surrounded by lipid and lipophilic surfactants from a hybrid structure. The particles showed a negative zeta potential with higher stability in different media, and the cellular uptake study showed an 83.3% particle uptake by L929 cells [55]. However, cellular internalization and intracellular drug release are two key indicators of the efficiency of a potential drug delivery system. Tahir et al. developed both hydrophilic and lipophilic doxorubicin (DOX)-loaded polymer core-shell structured hybrid particles for use in breast cancer [56]. The particles were synthesized via one-step nanoprecipitation with different formulations, where lipophilic DOX-loaded particles showed early uptake into the nucleus because of the binding between DOX and lipids in the cell membranes. The results indicated improved cellular uptake and better therapeutic efficacy of the chemotherapeutic delivered [56]. A few currently available passive targeted PLHNPs for cancer treatment are listed in Table 1. Besides the EPR effect, the physiology of the tumor microenvironment including high metabolic rate, higher oxygen level, high nutrient demand, and especially, the acidic nature of a tumor also facilitates passive targeting. Based on these advantages, many approaches have been developed for the effective delivery of anti-cancer agents to the tumor site. The procedures generally include the use of internal or external stimuli. Recently, Tran et al. reported the delivery of both docetaxel (DTX) and vorinostat (VRS) via a hybrid vehicle using an amphiphilic block copolymer that self-assembled into the NP with a hydrophobic PLA-poly (l-aspartic acid) core and further with a hydrophilic PEG-shell layer to form a CSPHNP [57]. Owing to the pH-sensitive properties of PLA, the hybrid system showed a pH-sensitive, controlled drug release because of the protonation of the carboxylic group in the tumor microenvironment. However, the results showed a higher release rate of VRS than DOX under equivalent acidic conditions [57]. However, the release rates of drugs were 42% and 25% less, respectively, compared with those observed in the study of Huang et al., who prepared a glutathione (GSH) sensitive prodrug carrier and showed an 85% drug release from the hybrid system when used against ovarian cancer [58]. The lipid-PLGA hybrid nanocarrier successfully delivered camptothecin (CPT), and resolved hydrophobicity in human ovarian cancer cells [59].

PLHNPs can be used as a versatile platform for small interfering RNA (siRNA) delivery through passive targeting. PLGA-PEG NPs incorporated with a cationic polymer/lipid synthesis form 1,2 epoxytetradecance by polymerization in the presence of dendrimers known as (G0-C14), which self-assemble with PLGA hydrophobic core. G0-C14 with a payload of cisplatin and siRNA can effectively downregulate target genes, both in vitro and in vivo, and exhibit multiple drug resistance-overcoming abilities [60]. Although modifying the size, shape, and a few surface dimensions of PLHNPs can regulate passive targeting, they have some limitations. Target cells within a tumor are not always conceivable as the drug diffuses to other off-target tissues or organs, and the random approach may affect the healthy tissue. As some tumors do not exhibit an EPR effect, the use of passive targeting is limited in biomedical applications. 

**Table 1 molecules-25-04377-t001:** Recently developed passive targeted anticancer therapies by PLHNPs.

Formulation		Delivery System	Therapeutic Cargo	Cancer Treatment	Activity	References
Polymer	Lipids					
Poly (lactic-co-glycolic acid) (PLGA)	Phosphatidylcholine and DSPE-PEG-2000	Polymer core lipid-shell hybrid systems (PCLHNPs)	Docetaxel, FTY-720 (SK1 inhibitor)	Metastatic prostate cancer	Overcome FTY70-induced lymphoma with higher toxic profile	[51]
Chitosan	Lipoid S75, DSPE-PEG	(PCLHNPs)	Cisplatin	Ovarian cancer	Prevent drug leakage by polymer, further by lipid layer with a safety profile	[50]
Cholic acid functionalized poly (dl-lactide)	Lecithin, DSPE-PEG	PCLHNPs	Paclitaxel, Celecoxib	Cervical cancer	Decreased the IL-10 cytokine production by drug resistance cell	[37]
PLGA	Lecithin, DSPE-PEG-2000	Polymer-Caged Liposome Hybrid Nanoparticles (PCLNPs)	Docetaxel	Cervical cancer	Deep tissue penetration and effective treatment compared to clinically available formulated drug	[53]
PEG-b-Poly (l-aspartic acid)	Caproyl 90, TPGS, DDAB	PCLHNPs	Docetaxel, Vorinostat	Breast cancer	Enhanced docetaxel activity	[57]
PLGA	DSPE-PEG-2000, Phosphatidylcholine,	PCLHNPs	Psoralen Doxorubicin (DOX)	Liver cancer	High physical stability for over 5 days	[61]
PLGA	Catanionic G0-C14, DSPE-PEG-2000	PCLHNPs	siRNA, Cisplatin	Breast cancer	90% gene encapsulation efficiency	[60]
PLGA	EPC (Egg phosphatidylcholine, DOPE (1,2-dioleoyl-sn-glycero-3-phosphovholine), PEG-2000	PCLHNPs	Camptothecin	Ovarian cancer	Resolved the water insolubility and sustained release of CPT inside the cell	[59]
PLGA	Soybean, lecithin, DSPE-PEG-5000	PCLNPs	Paclitaxel, Triptolide	Lung cancer	Higher tumor reduction from 1737 to 392 mm^3^	[62]
PLGA,	Soybean, lecithin, DSPE-PEG	Monolithic lipid-polymer hybrid NPs (MLPHNPs)	Psoralen Doxorubicin	Hepatocellular carcinoma	Enhanced Dox cytotoxicity via increased cytochrome c	[63]
PLGA	Cholesterol, DSPE-PEG	PCLHNPs	Docetaxel	Breast cancer	Enhanced pharmacokinetics	[64]
PLGA	Lipoid GmbH, DSPE-PEG	MLPHNPs	Paclitaxel, Etoposide	Osteosarcoma	Excellent tumor reduction and often 2-fold superior efficacy than free drug	[54]

## 4. Active Targeting with Surface Engineered PLHNPs

Active targeting is an advanced mechanism that enhances therapeutic effectiveness and decreases harmful side effects. It is a compelling way to deliver therapeutic drugs, genes, and imaging agents to the target site of interest without affecting healthy tissues and with better therapeutic efficacy and less toxicity [48]. Notably, active targeting can provide enhanced delivery compared with passive targeting because of target specificity and molecular recognition. In active targeting, the outermost surface of the NPs is coated with different types of targeting ligands, such as antibodies, peptides, aptamers, and small molecules [65]. These targeting moieties are directly attached to the surface of PLHNPs either by strong chemical conjugation or via electrostatic interactions. Active targeting depends on the biological interactions between the outer cells, targeting ligands, and the specific target cells. Currently, researchers are mainly focusing on active targeting strategies involving PLHNPs for effective treatment for cancer [51,56]. 

Receptor-mediated targeting entails the specific molecular targeting of certain overexpressed receptors in cancer cells. It is a well-established method to improve the nanocarrier accumulation within a specific tissue [66]. Mostly, via receptor-based targeting, a nanocarrier can be rapidly transported to the required target site with a high degree of accuracy. Thus, by combining the superiority of receptor overexpression with an uncommon binding motif, LPHNPs can be functionalized with different ligands to preferentially bind to the receptors via ligand-receptor interaction at the cell surface [67]. The ligand-engineered hybrid NPs are internalized by cancer cells via receptor-mediated endocytosis with a higher affinity towards the targeted tissues and a lower affinity towards the healthy tissues [68]. LPHNPs are functionalized by adding different ligands to the end of the PEG chain through covalent bond formation (amide-based reactions) or click chemistry. Thus, receptor-mediated targeting of PLHNPs enhances cellular uptake and cell reorganization by targeting ligands. Various surface receptors that are overexpressed on the cancer cell membrane are targeted to increase therapeutic efficacy. Currently, clinical trials are extensively based on active targeting drug delivery nanosystems [69]. Some of the receptors are discussed in the sections below.

### 4.1. Folate Receptors

Folate receptor (FR) is a cysteine-rich membrane glycoprotein with a high binding affinity [70,71]. This receptor is broadly used in the field of targeted anti-cancer therapy. FR is potentially highly expressed on various cancer cells, such as the breast, lung, brain, colorectal, gastric, prostate, and ovarian cancer cells [72,73]. The folic acid (FA) ligand-decorated PLHNPs allow active FR-mediated endocytosis with enhanced drug delivery to targeted cancer sites. The sorafenib (SRF)-loaded FR-targeted core-shell structured PLHNPs showed higher internalization, with killing lethal effect against FR-overexpressing liver cancer cells [74]. The core-shell hybrid NPs are formed using a chitosan and chondroitin sulphate-based natural polymer complex covered with a PEGylated lipid shell to improve stability. Further, the lipid shell is conjugated with FA for cancer cell targeting. The internalization study showed a stronger fluorescence intensity of core-shell hybrid NPs after 2 h compared with that of non-targeted particles, indicating a higher cellular uptake by FA. The quantitative analysis showed a higher uptake than that of the non-targeted particles. Based on the potential targeting ability, it exhibits sustained release of SRF with an efficient anti-cancer effect [74]. A similar study revealed that the folate ligand-modified novel PLHNP carrier was prepared using DSPE-PEG2000 lipid with a self-assembled poly (ε-caprolactone)-poly (ethylene glycol)-poly(ε-caprolactone) PCL-PEG-PCL copolymer core for targeted, controlled, and sustained delivery of paclitaxel (PTX) both in vitro and in vivo [75]. The NPs showed higher cellular uptake in FR-positive EMT6 cells (mouse carcinoma cells) than FR-negative L929 fibroblast cells, indicating that receptor-mediated endocytosis promotes the internalization of NPs into cells via FR-overexpressing EMT6 cell receptors. In vitro cytotoxicity assays demonstrated that the PTX-loaded ligand-coated particle had a 2-fold higher toxic effect than the non-functionalized particles. Besides, FA-functionalized particles showed a 65.78% tumor reduction, whereas the non-targeted particles showed a 48.38% tumor reduction [75]. These findings indicate that the lipid shells and biodegradable polymeric core contribute to an effective nano-drug formulation based on core-shell hybrid NPs coated with FA for targeted anti-cancer therapy.

PLHNPs not only display a high molecular-targeting ability but also provide stimuli-responsive drug delivery [76]. Thus, Gu et al. developed an efficient hybrid nanoparticle to overcome the limitations of cisplatin, including concentration-dependent aggregation, poor stability of ICG, and serious toxic side effects. The hybrid system contains a hydrophobic PLGA polymeric core that can carry both ICG and cisplatin in the lecithin monolayer of lipid, which improves the stability of the carrier and the last lipid shell for prolonged in vivo circulation and FA conjugation to enhance targeting ability. An FA ligand-functionalized hybrid structure measuring around 90–100 nm in diameter with negative zeta potential indicates extra stability and high monodispersion. The drug release showed a 60.03% burst release without laser by 72 h, but with laser, the release rate increased to 95.02% by 72 h [76]. In addition, a particle internalization study showed enhanced cellular uptake by FA-conjugated hybrid nanoparticles via receptor-mediated endocytosis by FR. However, the cytotoxicity was 5.30% and 7.05% of early apoptotic and necrotic cells, respectively, which indicates scope for improvement [76]. To further exploit their chemotherapeutic efficacy, the chemotherapeutic drugs were combined with additional cancer therapy, including targeted hybrid NPs. Recently, Yugui et al. synthesized core-shell PLHNPs loaded with FA-conjugated, gefitinib (EGFR inhibitor) and yttrium 90 (radioisotope) to treat nasopharyngeal cancer [77]. These FR-targeted core-shell PLHNPs showed synergistic chemoradiation therapy with enhanced in vivo anti-tumor efficacy and 90% drug release to the target site [77].

Many other PLHNP systems coated with FA have been developed for FR-coupled delivery of DOX. For example, Wu et al. have shown stable and highly monodispersed folate-targeted redox-sensitive release of DOX using the hybrid system via thiol-disulfide exchange reactions in the tumor microenvironment, where the hybrid system retained the advantages of the traditional hybrid system and exhibited additional active targeting and redox sensitivity [78]. The hybrid system showed selective targeting of KB cells compared with that of the FR deficient Cos-7 cells, with stronger fluorescence of DOX after 10 h. NPs are located on the cell membrane because of the folate-binding effect. All these studies suggest that the functionality of the folate moiety in PLHNPs enhanced drug delivery. 

### 4.2. Transferrin Receptors (TfRs)

Transferrin is a vital glycoprotein with certain advantages for targeted drug delivery. Transferrin can bind and transport iron into the cell via receptor-mediated endocytosis [79]. Compared to healthy cells, transferrin receptors are abundantly expressed in cancer cells [80]. Owing to the overexpression of TfRs, transferrin ligand can bind to TfRs, leading to targeted anticancer therapy [81]. Normally, transferrin conjugation is used to target different types of malignant cells [82]. PTX-loaded conjugated NPs exhibit three-fold higher cellular uptake with sustained release of the drug in human prostate cancer cell lines [83]. Similarly, Guo et al. have demonstrated the anti-proliferative activity of DOX through transferrin surface-modified lipid-coated PLGA NPs. TfRs are attached to the lipid shell via post-insertion method. A549 cells absorb this hybrid NP via TfR-mediated endocytosis, resulting in 2.8–4.1-fold higher targeting than the PLGA NP alone. However, the drug release data showed a 65% release rate within three days for hybrid structures compared with 95% for the polymeric NPs alone [84]. Further, to improve the therapeutic efficacy, recently, a new lipid-coated PLGA hybrid nanocarrier was synthesized by the solvent injection method [85]. This hybrid system resulted in a 36.6-fold encapsulation efficiency of an aromatase inhibitor (7α-(4′amino) phenylthiol-1,4-androstadiene-3,17-dione (7α-APTADD) and increased the therapeutic efficacy, and the transferrin protein-ligand was conjugated to the nanoparticle for selective targeting ability. The particle showed TfR-mediated endocytosis with marked cellular uptake by SKBR-3 cells (human breast cancer cell line) [85]. The sustained release of 7α-APTADD showed dose dependent aromatase inhibition with higher accumulation in the tumor. This study demonstrated the effective inhibition of aromatase inhibitors with selective targeting ability compared with that of a non-targeted hybrid [85]. To address the developed therapeutic efficacy of PLHNP, Wang et al. have designed a multifunctional PLHNP. They demonstrated biological effects with the redox-sensitive TF ligand modified afatinib (epidermal growth factor inhibitor)-loaded CSPLHNP. The system contains a PLGA core surrounded by cholesterol-PEG-COOH and an outer redox-sensitive Tf ligand. The disulfide bond was introduced into the ligand using cysteamine to form an amide bond and linked to the cholesterol-PEG-COOH again via an amide coupling reaction, as shown in Figure 5.

Afatinib was incorporated into the PLGA hydrophobic core based on π-π interaction, resulting in 90% encapsulation efficiency. The particle entered the cell via transferrin receptor-mediated endocytosis and targeted the plasma membrane and exhibited a controlled release of afatinib because of the GSH cleavable disulfide bond in the tumor microenvironment. In this study, the particle provided a longer systemic circulation time, GSH-sensitive drug release, and additional plasma retention effect with excellent tumor inhibition ranging from 919 mm^3^ to 212 mm^3^ in vivo in lung cancer [86]. The factionalized hybrid structures of polymer-lipid NP result in transfer of therapeutic drugs via selective targeting and exceptional therapeutic outcomes [84]. 

### 4.3. Cluster-of-Differentiation 44

Cluster-of-Differentiation 44 (CD44) is a well-known cell surface adhesion protein involved in cell-cell interaction, adhesion, cell migration, and cell homing. The cell adhesive and migration properties of CD44 are required for the preparation of the ERM (ezrin, radixin, moesin) protein complex. Hyaluronic acid (HA, also called hyaluronan or hyaluronate) is the major ligand for the activation of CD44 [87]. CD44 has multiple regions of cytoplasmic domain in the tail. These various isoforms enhance HA binding ability [87]. The breast cancer cells show a high expression of CD44. HA-modified PLHNPs carry both hydrophilic and hydrophobic drugs that can be efficiently delivered to the target sites by HA-receptor mediated endocytosis [88]. Because of the endosomal release of HA, the hybrid system exhibits high cellular uptake with controlled drug delivery to the breast cancer cells. Shao et al. have developed a dual drug-loaded PLHNP system to effectively overcome multidrug resistance in acute myeloid leukaemia (AML) [89]. The nanoparticle comprises a poly-ε-caprolactone polymer (PCL), lecithin lipoidal shell, and DSPE-PEG that are self-assembled via a one-step nanoprecipitation method. Further, the attached targeting ligand indicates multidrug resistance with effective AML therapy [72]. The targeting ligand is attached to the surface by the carboxylic group of HA and amine groups of DSPE-PEG via an amide bond. HA showed selective binding affinity towards the CD44 receptor and internalized by receptor-mediated endocytosis. The loaded DOX and gallic acid showed 80% in vitro release profile because of the cleavable hyaluronidase enzyme [89]. The gallic acid acted as a free radical and induced Ca^2+^ dependent apoptosis. Besides gallic acid, the DOX cloud efficiently inhibits the tumor size from 956 mm^3^ to 213 mm^3^, with a 77.7% inhibition rate [89]. Currently, one study reported the development of an HA-modified hybrid system for combination therapy of colorectal cancer. The use of irinotecan (chemo drug) and plasmid DNA in the carrier provides a combination of chemo-gene therapy. Irinotecan is loaded into the polymeric PLGA core via hydrophobic interactions, and the gene is loaded into the lipid shells [90]. The hybrid carrier enables over 80% drug and 90% gene-loading capacity. The in vitro cellular uptake study supports CD44 targeting of HA-modified LPNs, as shown in Figure 6. Because of the active targeting, the cytotoxicity result enhances cell inhibition efficacy in SW480 cells compared with that of unmodified particles. However, the toxicity profile of HUVECs showed no significant difference, which may be because of the higher toxicity due to HA binding affinity of the overexpressed CD44 receptors [90], as shown in Figure 6. Based on both targeting and delivery carrier of HA, a study was fabricated with hybrid structure with self-assembly of hyaluronic acid-paclitaxel (HA-PTX) with lysolipid- a thermosensitive lipid. The lipid shell carried a payload with marimastat (MATT), a matrix metalloproteinase inhibitor, to target lung metastasis. This study resulted in deep tissue penetration with 40–42 °C hyperthermia triggering drug release. The study established a multi-functional synthesis of hybrid carrier for hydrophilic drug with water soluble HA polymer and thermosensitive liposomes [91]. 

Nanovaccine is an emerging approach in many infectious diseases that induce a strong immune response. CD44 targeting PLHNPs have been studied to deliver vaccine inside the tumor. The binding ability of CD44-targeting DOTAP-PLGA nanovaccine induces an interesting cell-mediated immune response resulting in enormous cytotoxic damage [92]. This particle formulation also releases 80% ovalbumin (OVA) antigen to the target site [92]. The HA surface coated PLHNPs release the drugs and antigens from the NPs [89,92].

### 4.4. Epidermal Growth Factor Receptor

Epidermal growth factor receptor (EGFR) is a well-known tyrosine-kinase protein receptor that belongs to the family of epidermal growth factor receptors (ErbB). EGFR is a strong therapeutic target in many solid tumors including lung cancer, colorectal cancer, ovary, kidney, head and neck, prostate cancer, and especially breast cancer cells. It plays a decisive role in cell growth and differentiation [93,94]. Many researchers have validated EGFR-targeted anticancer therapy [95].

EGFR receptor-linked hybrid polymer-lipid nanosystems have been shown to improve receptor-mediated targeting. EGFR-targeted monoclonal antibodies are attached to the end of the PEG chain of hybrid drug carriers as targeting ligands, which results in an effective anticancer therapy. For example, an intelligent hybrid system was constructed, co-loaded with cisplatin and DOX via solvent extraction method [96], in which the EGF-modified-C/D-LPNs showed greater stability for 30 days and released 80% drug to the cancer sites [96]. The dual drug-loaded particles showed strong efficacy against lung carcinoma compared with that of the single drug-loaded particles [96]. Another study explained the therapeutic role of EGFR targeting hepatocellular carcinoma. NPs modified with cetuximab (an antibody used to target EGFR) were internalized via EGFR-mediated endocytosis and exhibited large tumor-homing potential because of the combination regimen [97].

HER-1 and HER-2 are also receptor protein kinases belonging to the ErbB family, which have attracted substantial attention because of their molecular targeting abilities. These receptors are remarkably overexpressed in many cancers, including breast, lung, and brain cancers. Cetuximab, erlotinib, and panitumumab are some of the well-known antibodies that are used to target EGFR [98]. A recent study demonstrated EGFR targeting of Caco-2 cells [99]. The use of afatinib (HER inhibitor) in the core-shell polymer-lipid hybrid NPs modulates apoptosis with a better inhibitory effect against colon cancer cells [99]. The formulation of PLHNP particles effectively inhibited the growth of multidrug resistant tumors.

### 4.5. Antibodies

Biological ligands have been identified and investigated to expedite active targeting of NPs [100]. Different types of ligands have been used in ligand-mediated targeting, including polysaccharides, peptides, small molecules, and proteins. These biological ligands often exhibit specific binding affinities for specific receptors on the targeted cells. The biological interaction between the target cell and ligands leads to better cellular uptake and cancer therapeutic efficacy [100]. Compared to other biological ligands, antibodies have wide applications in the field of nanomedicine owing to their active targeting ability [101]. Antibodies exhibit a strong binding affinity to antigens and cell surface receptors of cancer cells [101]. Many monoclonal antibodies (mAb) have been developed as anticancer agents with efficient targeting properties. These unnatural proteins act as natural antibodies in the immune system. Many monoclonal antibodies such as bevacizumab, cetuximab, rituximab, and ofatumumab, have been approved by the United States Food and Drug Administration (FDA), and some are under clinical consideration [102]. Targeted therapy with mAb-conjugated hybrid drug carriers is now popular. Hu et al. reported the benefit of anti-CEA (antibody against the carcinoembryonic antigen, a protein present in certain cancers)-conjugated PLHNPs against 90% of CEA-overexpressing pancreatic cancers. The antibody was linked to the lipid-containing maleimide outer layer via a maleimide-thiol coupling reaction. The targeted hybrid NPs showed high integrity and stability after internalization by the targeted BxPC and XAP-3 cells [103].

The mAbs can be synthesized in four different forms: murine (from mouse protein), chimeric (combination protein of mouse and human origin), humanized (a small part of mouse protein that is attached to the human protein), and purely human proteins [104]. These different types of engineered antibodies have higher affinity than normal antibodies [105]. Cetuximab is a chimeric antibody used to target EGFR to treat EGFR-overexpressing cancers. It is reported that cetuximab-modified NPs have multidrug loading capacity and enhanced therapeutic efficacy against cancers [97]. Zhang et al. designed a trastuzumab-bearing PLGA/PEI/lipid hybrid system to treat breast cancer via human EGFR-mediated targeting [106]. Antibodies and drug conjugates are generally used as personalized anticancer therapies owing to their targeting efficiency. In a previous study, the authors established the binding percentage of trastuzumab to the particle depending on the antibody ratio. The finding implies that the hybrid system can provide improved therapeutic potency when compared with the other groups because of its drug-antibody ratio and active targeting by trastuzumab [106]. However, the antibody fusion protein (sFVA) engineered by Yong et al. showed dual targeting strategy with LPHNPs towards leukemia cells as shown in Figure 7. The sFVA was non-covalently attached to the surface of DSPE-PEG lipid-biotin shell of the hybrid system and mesoporphyrin (SnMP), loaded into the PLGA core, resulted in targeted heme oxygenase-1 (HO-1) inhibition in acute myelogenous leukemia (AML) through SnMP. The antibody moiety actively targeted CD64+ leukemia cells and passively targeted CD11b+ based on the negative surface charge and phagocytosis. This study demonstrates a synergistic effect of combining chemo-sensitization and immune activation via daunorubicin (DNR)-responsive apoptosis and macrophage polarization [107].

### 4.6. Peptides

Currently, different types of peptides have been successfully used in targeted drug delivery for cancer treatment. Most peptides are not only used in targeting but also for cell penetration. Peptide ligands have diverse structures, origins, targets, and biomedical applications, and potential applications in targeted drug delivery. Arginyl-glycyl-aspartic acid (RGD) peptides have a strong binding affinity towards integrin receptors. This specific expression of integrin receptors has been used in active targeting. RGD peptides have shown remarkable delivery of anticancer agents and NPs into the tumor site by active recognition of the αβ-integrin receptor. Zhang et al. developed an iRGD peptide-modified co-delivery hybrid system to treat ovarian cancer, which overcomes MDR owing to the synergistic effects of tetrandrine (TET) and PTX, resulting in high tumor targeting with controlled drug release [108]. However, Zhang et al. have evaluated the synergistic effect of doxorubicin (DOX) and mitomycin C (MMC) with RGD-functionalized PLHNPs (RGD-DMPLN) to treat triple-negative breast cancer. The synergistic effect of DOX- and MMC-loaded particle often leads to the largest tumor accumulation via both tumor vasculature and integrin receptors. Intravenous injection of (10 mg/kg) RGD-DMPLN resulted in a 4.7-fold to 31-fold reduction in lung metastasis, which was measured via bioluminescence imaging and 2.4-fold to 4.0-fold reduction of area index in lung metastasis with a survival rate ranging between 35% and 57%. The RGD conjugated with hybrid NPs resulted in high tumor reduction and significantly inhibited lung metastasis with extended host survival [109]. In a similar study, the RGD-decorated PLHNPs were loaded with the combination of DOX and sorafenib (protein kinase inhibitor) to enhance the efficacy against hepatocellular carcinoma. The hybrid system was actively internalized into human liver cancer cells by integrin activation and resulted in high cytotoxicity and pro-apoptotic effect [110]. In 2018, to improve the targeted therapeutic efficacy, Wang et al. synthesized RGD-modified intelligent PLH nanocarriers against lung carcinoma with higher cellular internalization on A549 cell as shown in Figure 8. The RGD-ss-PTX/CDDP hybrid nanocarrier exhibited tremendous in vivo antitumor efficacy and suppressed the tumor size from 1486 mm^3^ to 263 mm^3^ [111]. Gao et al. formulated an integrin targeted PLHNP system to improve the therapeutic outcome of isoliquiritigenin (ISL) in breast cancer. This study showed that the targeted nanosystem was more effective than free ISL [112]. However, Li et al. reported a new strategy with a polymer-lipid-peptide (PLP) hybrid system to disrupt the tumor vasculature via depletion of tumor-associated platelets [113].

The hybrid system comprises poly(etherimide)-PLGA (PEI-PLGA) block copolymer core, which contains both DOX and R300 (an antiplatelet antibody), and a lipid shell layer containing lecithin, and a PEGylated phospholipid, which carry MMP2 cleavable peptide on the surface via chemical conjugation. The DDS facilitates onsite delivery of R300, which resulted in depletion of tumor-associated platelets via micro-aggregation. Systematic platelet depletion has been shown to enhance the therapeutic effect of chemotherapy drugs. Here, the combined effect of chemotherapeutic and antiplatelet depletion improved the treatment efficacy, reduced the toxic side effects, and resulted in a slow release of drugs due to the structural formulation of PLHNPs. Moreover, this antiplatelet depletion strategy represents a potential treatment and an advanced clinical strategy [113].

### 4.7. Aptamers

Aptamers are very short and artificial chemical antibodies that are developed from nucleic acids. They are highly sensitive, biocompatible, biodegradable, and poorly immunogenic, and thus compete for active targeting ligands [114]. Aptamers are single-strand oligonucleotides (DNA or RNA) that show high binding affinity to target proteins via specific reorganization. They have several advantages, such as facile synthesis, chemical modification, and high stability. Aptamer sequence can be modified for selective molecular targeting. One study investigated the targeting ability of MUC1 aptamer with drug loaded PLHNP to treat a malignant tumor. They conjugated MUC1 aptamer at different densities to the PEG layer via an amide bond and found increased cell targeting efficacy with an increase in density [115].

Gui et al. designed PLHNPs decorated with CD133 aptamers for targeted delivery of retinoic acid to osteosarcoma-initiating cells. CD133 is known as an osteosarcoma-initiating cell marker that is overexpressed in osteosarcoma cells. This study demonstrated the potential for targeted anticancer therapy of osteosarcoma-initiating cells as shown in the Figure 9 [116]. Aptamer-conjugated hybrid nanosystem was developed to target prostate-specific membrane antigen (PSMA) that is overexpressed in prostate cancer and resulted in a sustainable release of cisplatin [48].

Click chemistry is an efficient technique to synthesize aptamer-polymer hybrid structures (APHs) in a safe and effective manner. Oh et al. synthesized a hybrid structure of a cell targeting aptamer (nucleophilic-specific aptamer) coupled with block copolymers to target tumor cells [94]. APHs were used for cell-specific targeting and internalized into the target cells through active endocytosis [117].

### 4.8. Dual-Targeting Ligands

The targeting of two different types of receptors of cancer cells via dual ligand-modified NPs represents an advanced strategy in the field of nanomedicine. Dual-targeting ligand-decorated carriers have garnered growing attention from many scientists. The concept of dual targeting has been applied in some nanosystems but is still not widely used because of the interplay of numerous factors such as the choice of ligands, ratio, density, size, and matching of dual ligands. The use of dual-targeting ligands in nanosystem to enhance cellular uptake by cell-specific targets of both ligands results in higher accumulation in tumors. Yang et al. investigated the use of dual-targeting ligand-modified PLHNP to treat breast cancer. They used anti-HER2 and neu peptides modified with HIV-1 Tat (m TAT) for targeted delivery of DTX in HER2/neu-overexpressing cells. NPs are formulated with a polymeric core covered 90% with lipid bilayers and a 5.7 nm hydrated PEGylated shell. The dual-targeting ligand-decorated PLHNP system carried hydrophobic drugs with controlled, targeted delivery to the target site and represented an attractive vehicle for targeted anticancer treatment [118]. A similar study targeted aggressive triple-negative breast cancer in 2017 in a lung metastasis model to investigate the dual-targeted anticancer effect of DOX, NAD, and mitomycin C (MMC)-co-loaded PLHNP. The hybrid system with an optimal density of RGD exhibited a 31-fold reduction in the lung metastatic load with 57% extended survival rates [109].

### 4.9. Small Molecules

Small molecules act as ligands in targeted anticancer therapy and are targeted towards specific biological macromolecules. Small molecules are low-molecular-weight organic compounds of 1 nm size. Many drugs behave as small molecules. Small-structured amino acids, monosaccharides, RNAs, proteins, and nucleic acids are considered small molecule therapeutic agents for anticancer treatment. For example, FRs represent targeting receptors for FA, which is a certified small molecule. FA-conjugated lipid monolayer and polymeric core hybrid NPs were developed to deliver docetaxel (DTX) to target cells [29]. Another study purposed to treat nasopharyngeal cancer with an FA-grafted dual drug-loaded core-shell structure of PLHNP. The hybrid system simultaneously inhibited epidermal growth factor (EGF) with combined radiotherapy using Gefitinib and Y90. The nanosystem was internalized by receptor-mediated endocytosis and showed improved NPC cell uptake with 90% drug release without any systemic toxicity [77]. Phenylboronic acid (PBA) is another example of a small molecule targeting ligand. Deshayes et al. showed that PBA binds to *N*-acetylneuraminic acids, which are the main ingredients of sialic acid (SA). It is reported that most SA is present on the surface of cancer cells [119]. Recently, a dual-sensitive PLHNP was fabricated as an anticancer drug carrier, which has pH and redox-sensitive release profile at the target site. The hybrid system shows enhanced serum stability and 90.2% sustained drug release. This study demonstrated the therapeutic efficacy of small molecule-grafted hybrid nanomedicine. However, further improvement is necessary for better therapeutic effectiveness. Currently, researchers are exploring additional opportunities in the area of chemistry, biology, and medicine by targeting RNA with small molecules [120]. Small molecules and RNA interaction can facilitate the discovery of new drugs in cancer treatment [120]. Currently used active targeted PLHNPs for cancer treatment are listed in Table 2.

## 5. Applications of Polymer-Lipid Hybrid Nanoparticles

### 5.1. Drug Delivery

PLHNPs have wide applications in cancer treatment. They can efficiently encapsulate and deliver either single or multiple therapeutic drugs to the target sites with distinct types of loading methods. Additionally, PLHNPs have gained enormous attention because of the efficient delivery of nucleic acids, genes, siRNAs, DNA, vaccines, and imaging agents for cancer therapy. Compared to traditional nanoparticles, PLHNPs have shown great potential in terms of drug-entrapment efficiency, cellular internalization, and drug release kinetics. PLHNPs are differentiated into three classes, including single drug delivery, multiple drug delivery, and actively targeted drug delivery with different surface-coated ligands. It has been shown that the nanoparticle size and shape determine the drug delivery performance. Thus, the PLHNP size and surface charge are crucial for drug delivery to prevent particle aggregation, stability, and pharmacological drug effect [127]. The particle size and surface charge of cisplatin-loaded chitosan-lipid hybrid NPs formulated with different ratios were measured. A ratio of 20:1 had excellent stability with 70% drug-entrapment efficiency [50]. The 173 nm PLHNPs often delivered both hydrophilic and lipophilic DOX to the cancer site [56]. It is also reported that the ultra-small PLHNPs (25 nm) localized in a deep tissue region of the tumor with effective PTX delivery and excellent in vivo and in vitro stability in a mouse tumor model [53]. PLHNPs facilitate the delivery of numerous single chemotherapy drugs such as temozolomide, docetaxel, salinomycin, triptolide, paclitaxel, doxorubicin, and hydroxycamptothecin, among others specifically targeting different types of cancer [62,128,129]. Single drug-loaded PLHNPs have reduced in vivo and in vitro toxicity, good biocompatibility, excellent cellular internalization, and enhanced release kinetics [26,50,83]. Compared to single-drug chemotherapy, dual or multidrug therapy is more effective in overcoming MDR. PLHNPs have been used widely as multidrug carriers for combination therapy of cancer. The combination of paclitaxel and tetrandrine has been used to treat cancer. The dual drug-loaded particles show distinct controlled release for both chemotherapeutic drugs [89]. The combined drug delivery results in targeted delivery via different surface-decorated ligands already discussed above. A targeted PLHNP system has been developed to deliver both the chemotherapeutic drug and radiotherapy agents for cancer treatment [77]. Active targeted treatment reduces the toxicity and delivers the drug towards cancer sites [76]. FA surface-decorated and c(RGD)-conjugated PTX hybrid exhibited dual-targeting delivery to the glioma cells [130]. This intelligent carrier effectively bypassed the blood-brain barrier and resulted in targeted delivery using both drugs [130].

In cancer treatment, stimuli-responsive PLHNPs were effective because of their enhanced cellular internalization, excellent distribution, and well-controlled stimuli-responsive release with specific delivery to the target site. Core-shell PLHNP surface decorated with RGD peptide and PTX prodrug linked to a redox-sensitive bridge (RGD-ss-PTX) resulted in active targeting via integrin receptors and GSH-responsive drug release with improved antitumor efficacy in the lung cancer xenograft model [111]. Redox-pH dual responsive PLHNPs fabricated by self-assembly of amphiphilic polymer and methyl-ethyl grafted disulfide bond contained poly (β-amino esters) and further organized according to the PEGylated lipid layer. The hybrid nanosystem showed pH-sensitive controlled drug delivery at pH 6.55 and effectively targeted tumor cells compared to the free DOX group [131]. In addition to the stimuli-responsive drug delivery, the lipid polymer hybrid NPs efficiently deliver photosensitizer and a photothermal dye resulting in a synergistic photodynamic and photothermal therapy. Diaye et al. designed a chemotherapeutic drug beta-lapachone (β-Lap) and photosensitizer m-THPC co-loaded in a different area of the biodegradable PLHNP system for the treatment of retinoblastoma malignant tumor. This hybrid carrier can be used for combined chemo-phototherapy in both oxygenated and hypoxia tumor tissues via single IV injection and concurrent phototherapy induced the expression of NQO1 in Y79 cells, offering a better therapeutic strategy for malignant tumors [132]. Another study reported improved therapeutic efficacy of combined photothermal-chemotherapy by polyaniline-lipid hybrid NPs. The polyaniline acted as a fluorescence resonance energy receptor (FRET) to efficiently convert the energy of near-infrared fluorescence (NIRF) from the donor, which is an attractive strategy to improve the efficacy of photothermal therapy (PTT) with a controlled drug release profile [133]. Similar to external stimuli such as lasers, the treatment with PLHNPs also improved with ultrasound. Huang et al. reported the synthesis of a hybrid system comprising PLGA and DSPE-PEG-cRGD, which contain perflurohexane (PFH) in the core liquid and GSH-sensitive platinum drug in the lipid shell. The hybrid system often facilitated ultrasound-induced drug release inside the tumor cell, resulting in GSH depletion and enhanced synthesis of reactive oxygen species (ROS), which triggered mitochondria-mediated apoptosis as shown in the Figure 10 [58].

### 5.2. Gene Therapy

Gene therapy is an alternative method to treat cancer by delivering therapeutic genes (nucleic acid) instead of chemotherapy drugs. It is the latest advance of research interest because of the genetic modifications. Among traditional vehicles such as biodegradable NPs and cationic liposomes, PLHNP carriers also have applications in gene therapy. Because of high stability, long shelf-life, and biodegradability, PLHNPs have emerged as gene delivery vehicles for cancer. Small interfering RNA (siRNA) is an emerging agent for the treatment of numerous diseases. Gao et al. reported a siRNA mixed core-shell type lipid/rPAA-chol polymer nanocomplex for transferrin receptor-targeted in vivo siRNA delivery. The synthesized rPAA polymers attract negative siRNA by their positive charge via electrostatic interaction. The nanoparticle is internalized by receptor-mediated endocytosis and the siRNA is released into the cytoplasm. This study resulted in greater gene silencing with enhanced in vivo tumor inhibition [134]. The self-assembly of cisplatin and DNA entrapped in the cationic lipid-polymer hybrid NPs was studied in the treatment of pediatric head and neck cancer. The CDDP/DNA core-shell NP exhibited enhanced in vitro transfection efficiency compared to the naked DNA. This smart nanoparticle loaded with both drug and gene carriers enhances in vivo antitumor efficacy in RD-4 tumor-bearing BALB/c mouse model [135]. However, microfluidic synthesis of PLHNPs resulted in excellent stability and safe delivery of siRNA as shown in Figure 11. The hybrid system coated with neutral lipid layer over polyethyleneimine-polycaprolactone (PEI-PCL) polymeric micelle resulted in reverse micelle formation inside the tissue. The study demonstrated the EGFR downregulation of protein both in vitro and in vivo [136].

It is reported that siRNA is rapidly degraded in the nucleus because of its short half-life [137]. To resolve this issue, Zhao et al. employed biocompatible PLHNPs containing cationic polylysine co-polymer and PEGylated lipid layer. The lipid bilayer shell carried the siRNA inside the nanoparticle. The short life of siRNA was increased and resulted in tumor eradication in the orthotopic tumor model [138]. All the above studies demonstrate that PLHNPs are appropriate vectors for drug delivery because of their stability, long life, and biodegradable properties [68].

### 5.3. Delivery of Imaging Agent

PLHNPs can improve the performance of both therapeutic drugs and imaging agents for enhanced medicinal diagnosis. The imaging agents allow the visibility of the target tissue. Imaging can be used to detect the early stages of cancer and rapidly decide the treatment options [139]. Various types of imaging agents, including fluorescent dyes, quantum dots, iron oxides, and inorganic nanocrystals are commonly used in molecular imaging modalities such as magnetic resonance imaging (MRI), computed tomography (CT), and photoacoustic (PA) imaging [139,140]. Wo et al. have reported FR-targeted PLHNPs coated with gadolinium (Gd-FPLHNPs) for imaging and targeted DOX delivery [141]. Gadolinium and DOX incorporate into the lipid and PLGA layers of the NPs. The active targeted Gd-FPLHNP showed high paramagnetic properties with 2-fold enhanced longitudinal relaxivity and enhanced cellular internalization by folate, and higher cytotoxicity in vitro in epidermoid carcinoma [141]. Another study reported the use of QD with the hydrophobic polymer to formulate QD-PLHNPs for improved stability with enhanced fluorescence imaging [142]. A study demonstrated the ultrasound triggered satisfactory release of DOX with image-guided tumor therapy by using a multiparous polymer-lipid (lipid-PLGA) hybrid NPs as shown in the Figure 12. 

Apart from imaging agents, fluorescent dyes have been loaded onto the PEGylated lipid shell to form hybrid NPs. Such particles exhibit brilliant fluorescence imaging capabilities [144]. Some Au nanocrystal loaded PLHNPs also facilitated in vitro bioimaging, demonstrating higher CT and visual imaging. Similarly, manganese (Mn) nanoparticles are well known for their tumor hypoxia modulation in MRI applications. Gordijo et al. designed a hybrid system with a MnO_2_ nanoparticle with a hydrophobic terpolymer-lipid matrix. The Mn-embedded hybrid system resulted in significantly high colloidal properties and biocompatibility both in vitro and in vivo. The use of Mn reduced tumor hypoxia in solid tumors under both local and systemic administration. The hybrid system additionally prevented premature Mn loading and systemic oxygen generation in tumor microenvironment [145]. These findings demonstrate the selective imaging ability of PLHNPs in cancer therapeutics.

### 5.4. Immunotherapy

In the last several years, immunotherapy has played a prominent role in the treatment of cancer by boosting the body’s natural immune mechanism. Some recent studies have reported the development of PLHNPs in cancer immunotherapy to deliver vaccines and immunotargeting of drugs to the target side. Liu et al. used a cationic lipid-poly(lactic-co-glycoside) acid and PLGA polymer hybrid nanoparticle for antigen delivery to the cancer cells. They formulated three types of OVA antigen-absorbed or -encapsulated hybrid particles to investigate antigen-specific responses. The antigen-loaded NPs facilitated efficient delivery into the draining lymph node with higher CD86 expression compared with free OVA. The study outcomes demonstrated that antigen availability plays an important role in immunotherapy. In addition, both encapsulated antigen NPs exhibited efficient dendritic cell (DC) activation and induced high expression of MHCs and T cell activation [146]. However, Zhang et al. improved DC targeting via PLHNPs using PCL-PEG-PCL polymer, cationic lipid 1,2, -dioleoyl-3-triethylammonium propane (DOTAP), and the DSPE-PEG-mannose-containing hybrid formulation. The system often contains a high load of TLR4 agonist monophosphoryl lipid A (MPLA) in the lipid layer and TLR7/8 agonist imiquimod (IMQ) in the polymer core. The OVA antigen was incorporated into the cationic lipid surface via electrostatic interaction and the mannose moiety in the PEGylated lipid. This intelligent hybrid formulation provides greater cellular internalization with 90% DC targeting efficacy after 48 h, which demonstrated particle biocompatibity [147]. Meanwhile, the core structured hybrid particle showed a release profile of 90% OVA and 85% IMQ. The antigen cross-presentation of DC leads to MHC molecule presentation to active CD8+T cells (T lymphocytes) to elicit enhanced cytotoxic T lymphocyte (CTL) response, as shown in Figure 13. The synergistic effect of Toll-Like Receptor (TLR) agonist triggered a strong innate and adaptive immune response via activation of both T and B cells [147].

Regulatory T cells (Treg) play a crucial role in preventing autoimmunity. Recently, a PLHNP was designed to target Treg in the tumor microenvironment to achieve an effective immune response. The nanoparticle was decorated with tLypl peptide to target Nrpl receptors on Treg cells. The PLGA polymer core also contained imatinib (a tyrosine kinase inhibitor) to block the STAT3 and STAT5 signaling pathway. The hybrid NPs ensured uniform distribution in the tumor microenvironment both in vivo and in vitro. Further, the enhanced targeting was mediated via the Nrp1 receptor and the activation of CD8+ T cells was induced using a CTLA4 checkpoint inhibitor. Thus, PLHNPs provide targeted delivery by enhancing anti-tumor immunity via synthesis of IFN-γ and TNF-α and suppression of Treg cells. All treatment paradigms demonstrated the induction of a significant immune response by PLHNPs compared with that of the control groups [148]. Thus, the use of PLHNPs with vaccine and CTL4 antibodies is a promising means to negate tumor growth and elicit a strong anti-tumor immune response.

## 6. Clinical Studies

The initial clinical success of both liposomes and polymeric NPs is attributed to their beneficial characteristics. Many liposomal drugs are currently available commercially, and some are under clinical trial. DaunoXome (daunorubicin liposomes), ambisome (amphotericin B liposomes), myocet, and mepact are examples of available liposomal drugs [19]. However, the use of polymeric NPs in clinical studies still needs to be explored. The widespread clinical use of PLHNPs can overcome the limitations of both liposomal and polymeric NPs. Many hybrid particles have recently been submitted for approval for pre-clinical study. In a preclinical study for breast cancer using targeted PLHNPs, Jain et al. investigated the targeting potential of fructose-modified beta carotene (BC) and methotrexate (MTX)-co-loaded PLHNPs and found a maximum apoptosis index of 0.89 against MCF-7 breast cancer cells. In addition, the loaded beta carotene and MTX induced hepatic and renal toxicity in the pharmacodynamic animal model [149], suggesting a potential therapeutic role, which has yet to be explored in preclinical research. Similar to most of the clinical studies based on oral drug delivery, many PLHNPs intended for oral drug delivery have yet to clear preclinal trials. Their toxicity barriers and safety profiles remain to be established before clinical studies are conducted. PLHNPs represent robust DDS and show enhanced cellular uptake. The pharmacological and toxicological aspects of these NPs need to be elucidated to determine their clinical impact.

## 7. Summary, Future Perspectives, and Challenges

As evident from our discussion, PLHNPs have been developed as theragnostics and present numerous advantages compared with both liposomes and polymeric NPs. They exhibit improved biocompatibility and stability, elongated circulation time, specific targeting ability, improved drug-entrapment efficiency with controlled release, and a high potential to improve the therapeutic efficacy in MDR and for other therapeutics. All these features suggest considerable potential of these NPs for cancer therapy. Despite the progress in the synthesis, characterization, surface decoration, and application of PLHNPs, key challenges remain to be addressed before using them as robust DDS for biomedical applications in cancer.

The density of the targeting ligand on the NP surface is one of the leading challenges limiting their high therapeutic potency. It is well known that the physiological properties of PLHNPs, such as the surface charge, particle size, and PEG chain density strongly affect their in vivo pharmacokinetic features. The functionalization of targeting ligands thus enhances not only the specific targeting ability via cell-tissue interaction, but also the surface properties of PLHNPs with consequent profound effects on in vivo pharmacokinetics. In Section 5 of this review, we discussed several ligands targeting the surface of hybrid NPs leading to high tumor accumulation. The optimization of ligand density via in vivo experiments to improve the pharmacokinetic properties of NPs is thus an intriguing research prospect. The second challenge is the explicit control of multiple drugs with distinct hydrophobicities inside the hybrid NPs. Many studies have demonstrated the potential of dual drug loaded PLHNPs. Under these circumstances, it is very important to control the molar ratio of the two drugs and their loading yield percentage, which suggests the possibility of future use of multi-drug loaded PLHNPs to elicit synergistic effects. A third challenge is the use of biomimetic PLHNPs. Cancer cell membranes coated with PLHNPs exhibit considerable potential owing to their homogeneous targeting capability. However, challenges include the retention of membrane properties, membrane fluidity, prolonged circulation times, and better preparation methods for PLHNPs, all of which can be addressed by formulating the polymeric core with nanogels. Cell membrane fluidity can be enhanced by the core of the hybrid NPs as in protected shells [150]. The core shell will increase the desirability of NPs. Thus, the fluidity of the cell membrane coated with nanohybrid system must be improved.

Finally, the large-scale production of these hybrid nanoplatforms has garnered considerable attention, which may represent an important aspect regulating translational research into these hybrid drug carriers. The integrity of these formulations can change dramatically during the scaled production of PLHNPs economically. However, the translation of PLHNPs into clinical applications is still in its infancy stage. The stability, toxicity, safety, and pharmacokinetics properties are the key aspects that should be focused on in addition to the regulatory strategies necessary for predicting the potential risk and assessments of PLHNPs in the market. For example, there is a need for proper selection of solvent, material, and procedure before NPs fabrication. Specifically, the solvent used during synthesis can affect the stability of the NPs. During scaling-up, they may induce toxic problems for the environment. In addition, during scaling up of PLHNPs, stability, binding ability, and circulating property could be compromised. To address these, there is a need to focus on developing solvent-free and easy scaling-up preparation methods suitable for the fabrication of PLHNPs. One of the major obstacles for the commercialization of PLHNPs is the need for expensive clinical studies, which could be overcome by investments from pharmaceutical and medical device corporations. Overall, the optimization of therapeutic requires easy production steps, cost, scale-up, and the addressing the regulatory concern can assist PLHNPs to attain the potential clinical applications.

In future, the progression of these hybrid NPs for cancer treatment may facilitate clinical studies, because this hybrid formulation contains clinically approved materials with superior stability and biocompatibility compared to those of traditional vehicles. PLHNPs exhibit highly desirable targeted delivery, tumor accumulation, and deep penetration owing to their small size and surface functionalization with different targeting ligands. In addition, they carry combinations of drugs, genes, vaccines, and imaging agents to facilitate effective treatment for cancer. They can also reduce degradation and shield the drug molecules owing to the outer PEGylated lipid layer. The unique properties of these NPs ensure a wide range of biomedical applications, such as anticancer therapy, immunotherapy, inflammation treatment, and bioimaging. Thus, PLHNPs represent a smart and attractive nanoplatform for anticancer therapy and may be the focus of future clinical investigations.

## Figures and Tables

**Figure 1 molecules-25-04377-f001:**
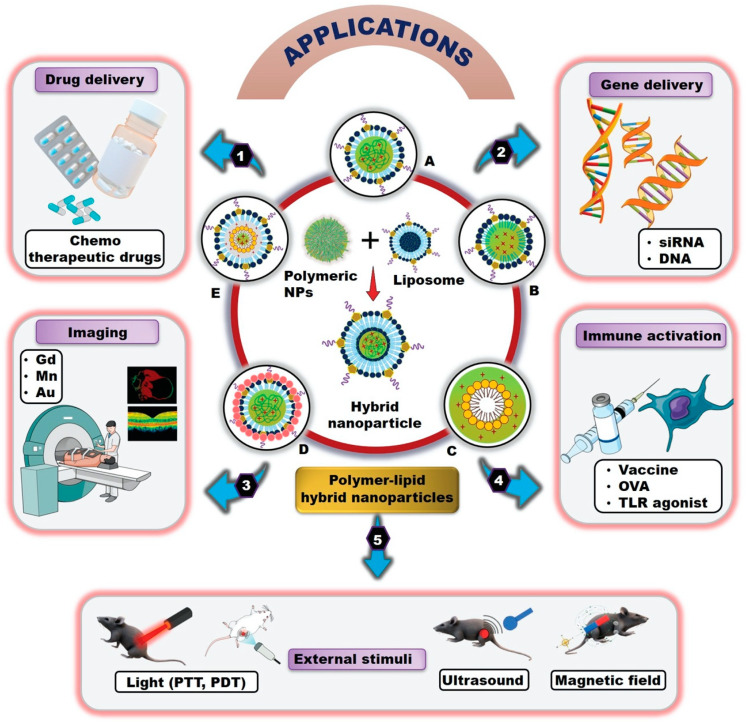
Overall schematic representation of polymer-lipid hybrid nanoparticles (PLHNPs) and their applications. The hybrid system is developed from polymeric nanoparticles and liposomes covered by different types of PLHNPs. (**A**) Polymer core lipid shell, (**B**) Polymer caged liposome, (**C**) Monolithic PLHNPs, (**D**) Erythrocyte membrane-coated PLHNPs), (**E**) Core shell-type hollow lipid-polymer-lipid hybrid nanoparticles. The different applications of PLHNPs are also shown. (**1**) Delivery of various chemotherapeutic drugs. (**2**) Delivery of genes (siRNA, DNA). (**3**) Delivery of vaccines, ovalbumin (OVA), and Toll-Like Receptor (TLR) agonist for immune activation. (**4**) Imaging applications based on gadolinium (Gd), manganese (Mn), and gold (Au). (**5**) Photothermal therapy (PTT), photodynamic therapy (PDT), ultrasound, and alternative magnetic field (AMF).

**Figure 2 molecules-25-04377-f002:**
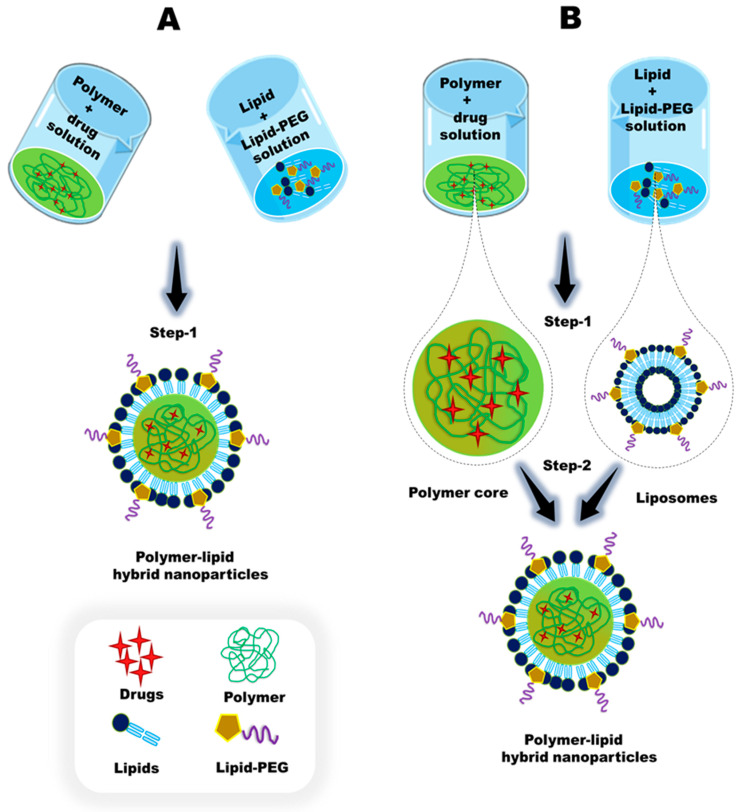
Synthesis of polymer-lipid hybrid nanoparticles. (**A**) One-step synthesis of PLHNPs by mixing drug-containing polymeric solution and aqueous lipid solutions. (**B**) Two-step synthesis of PLHNPs by mixing liposomes with the prepared polymeric drug solutions.

**Figure 3 molecules-25-04377-f003:**
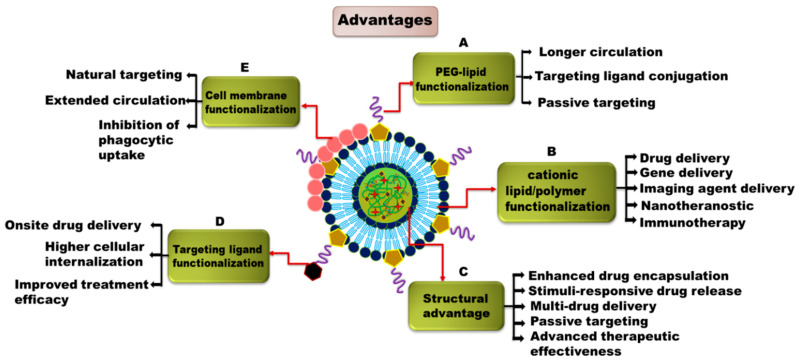
Schematic representation of advantages of different PLHNPs. (**A**) Polyethylene glycol (PEG)-lipid functionalization of PLHNPs prolongs the systemic circulation in the reticuloendothelial system (RES), inhibition of phagocytic uptake, and polymer tailing for the functionalization of targeting ligands. (**B**) The cationic lipid/polymer functionalization enhances the delivery of any drug, gene, vaccine, or imaging agent into the cells, resulting in theranostics applications against cancer. (**C**) A variety of PLHNPs facilitate multidrug encapsulation with drug release induced by different polymers and lipids, passive targeting by PEG polymer, and advanced therapeutic effectiveness of the hybrid system. (**D**) The surface fictionalizations of different target ligands enhance the cellular uptake and therapeutic efficacy as well as site-specific drug delivery. (**E**) The cell membrane coating of PLHNPs provides a natural homotypic targeting with extended circulation by simulating the membrane and inhibiting the phagocytic uptake by immune cells.

**Figure 4 molecules-25-04377-f004:**
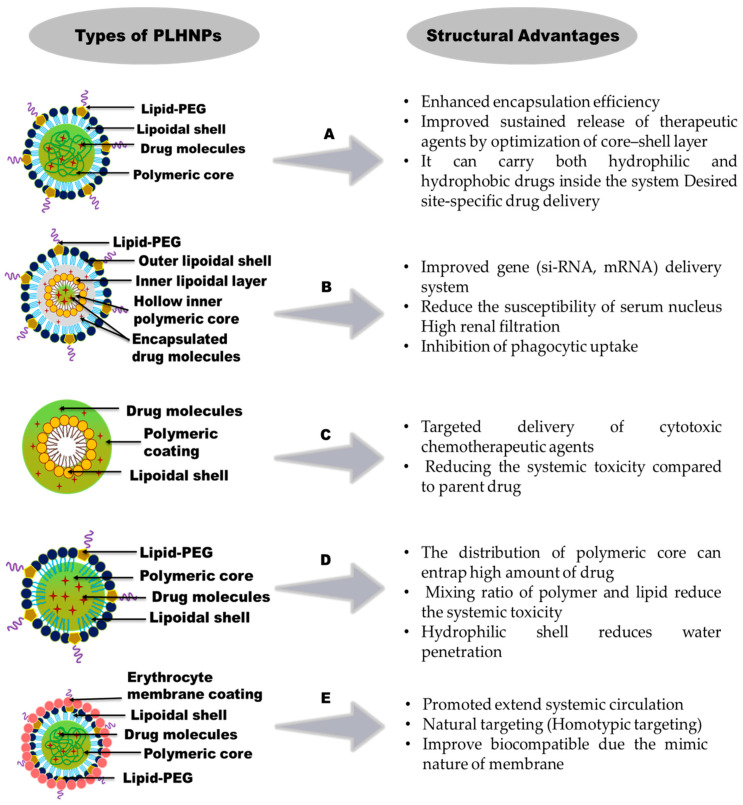
Different types of polymer-lipid hybrid nanoparticles and their structural advantages. (**A**) Polymer core lipid shell. (**B**) Core shell-type hollow lipid-polymer-lipid hybrid nanoparticles. (**C**) Monolithic PLHNPs. (**D**) Polymer-caged liposome. (**E**) Erythrocyte membrane coated PLHNPs.

**Figure 5 molecules-25-04377-f005:**
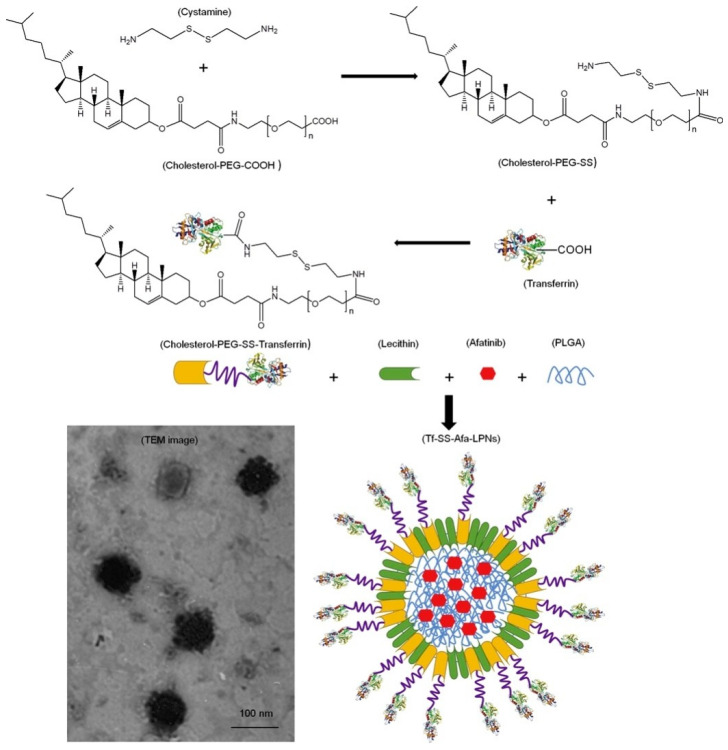
Synthesis of Cholesterol-PEG-SS-Transferrin and the transmission electron microscopy (TEM) image of afatinib-loaded cho-PEG-ss-Tf PLHNPs. The particle was synthesized by conjugating cholesterol-PEG and transferrin with cysteamine via amide bond formation, followed by afatinib (a tyrokinase inhibitor) loaded into the PLGA core through hydrophobic interactions. Transferrin protein binds to the transferrin receptor at the cell membrane and enhances drug delivery in the tumor cells and resulted in glutathione (GSH) stimuli responsive (disulphide bond breakage in the presence of GSH) drug release. Reprinted with permission from [86], Copyright © 2019 under American Chemical Society.

**Figure 6 molecules-25-04377-f006:**
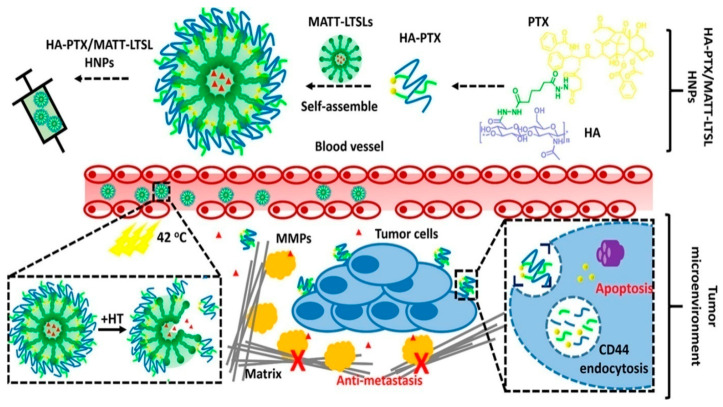
Synthesis of multifunctional LTSL (Lysolipid-containing thermosensitive liposomes) targeting CD44 receptor. The hyaluronic acid-paclitaxel (HA-PTX)-self-assembled, conjugated thermosensitive liposomes carried a large payload of marimastat (MATT) in the lipid shell with efficient MMP enzymatic inhibition of lung metastasis. Reprinted with permission from [91], Copyright © 2018 under American Chemical Society.

**Figure 7 molecules-25-04377-f007:**
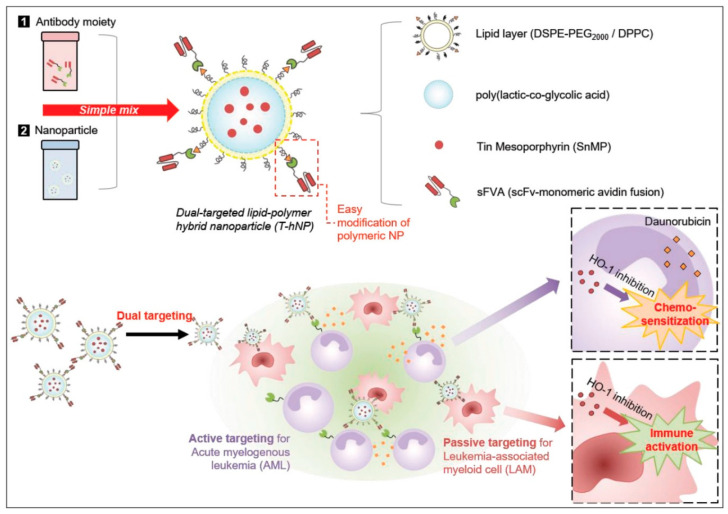
Dual targeted intelligent PLHNPs. The antibody-attached and SnMP (inhibitor)-loaded hybrid system exhibit a synergistic effect of chemo-sensitization and immune activation through hemeoxygenase-1 (HO-1) inhibition in acute myelogenous leukaemia cells. Reproduced with copyright permission from [107], under Creative Common Attribution 4.0 International License (CC.BY license).

**Figure 8 molecules-25-04377-f008:**
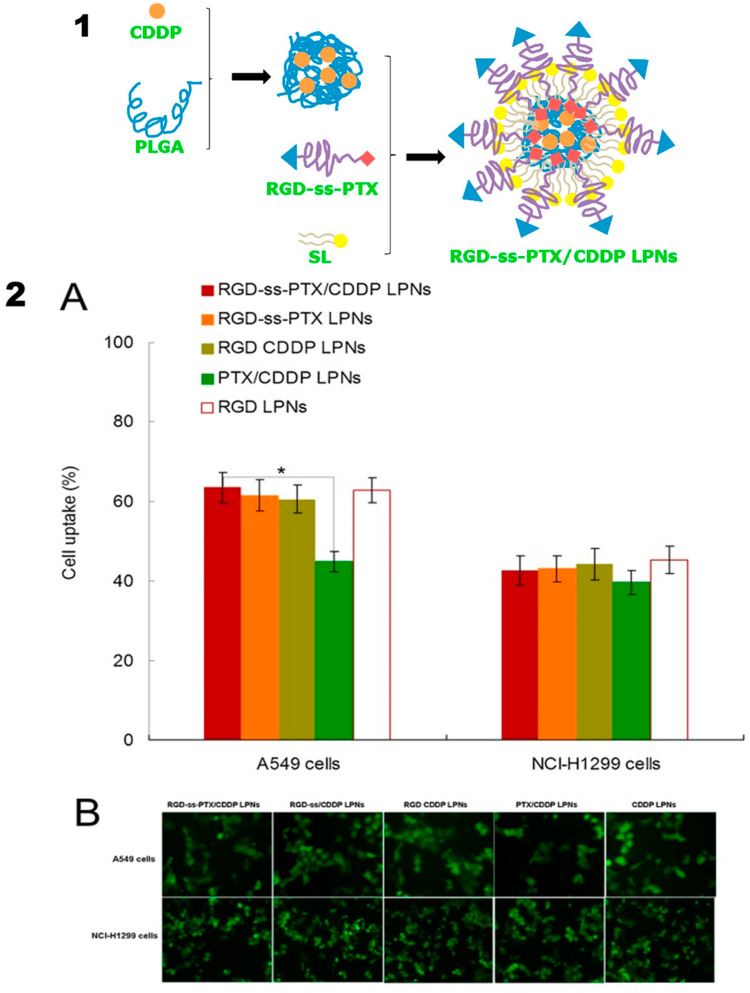
(**1**) Synthesis of drug delivery system (DDS) based on redox-responsive polymer-lipid-peptide hybrid NPs based to deplete tumor against lung carcinoma. (**2A**) The hybrid system is loaded with cisplatin and paclitaxel, which results in a synergistic effect of the chemotherapeutic agent treatment with high cellular uptake in the lung cancer, data represent mean ±SD (*n* = 6), * *p* < 0.05, and (**2B**) Fluorescence images resulted RGD-modified LPHNPs exhibit stronger fluorescence than non-ligand modified hybrid NPs. Reprinted with permission from [111], Copyright © 2018 Elsevier Masson SAS.

**Figure 9 molecules-25-04377-f009:**
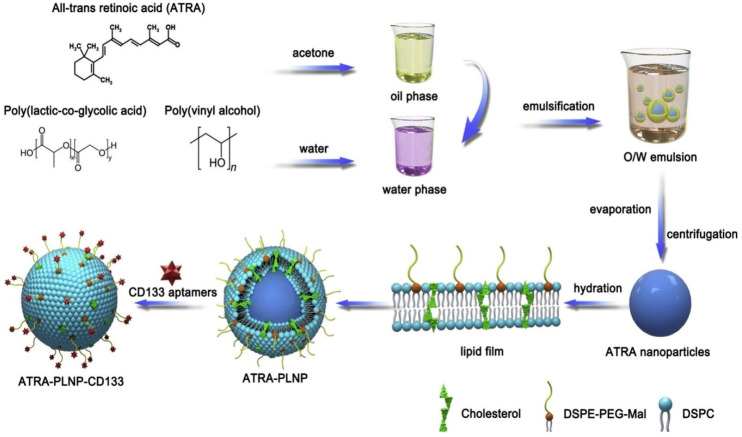
The synthesis of CD133 aptamer-conjugated all-trans-retinoic acid (ATRA) carrying PLHNPs. Reproduced with permission from [116], under Creative Common Attribution 4.0 International License (CC.BY license).

**Figure 10 molecules-25-04377-f010:**
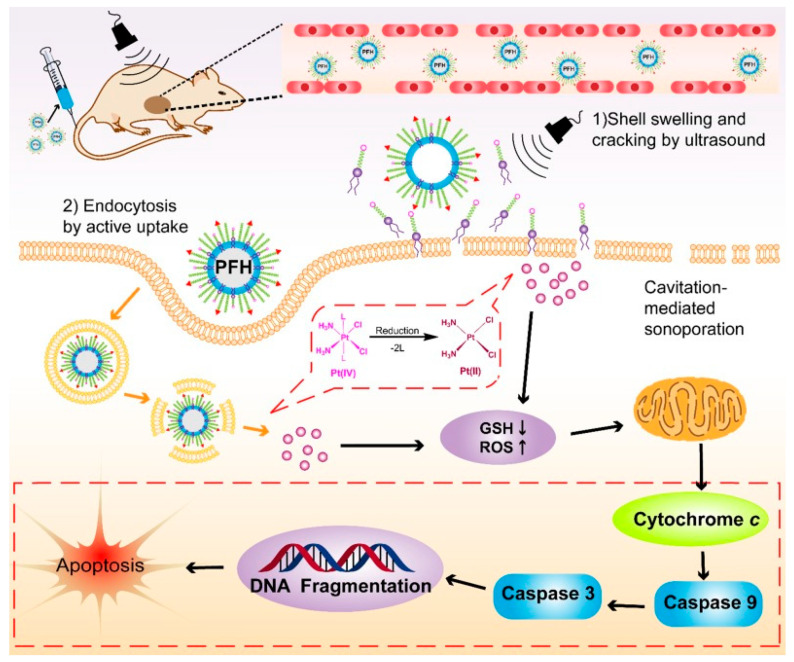
A hybrid polymer-lipid shell representing the targeted GSH-sensitive, ultrasound-triggered DDS. The hybrid system comprises a perfluorohexane (PFH) liquid core and platinum prodrug inside the lipid layer. The hybrid DDS often emits echogenic signals and enhanced therapeutic treatment via increased reactive oxygen species (ROS) level. Reproduced with the permission from [58] under Creative Commons Attribution 4.0 International License (CC.BY-NC license).

**Figure 11 molecules-25-04377-f011:**
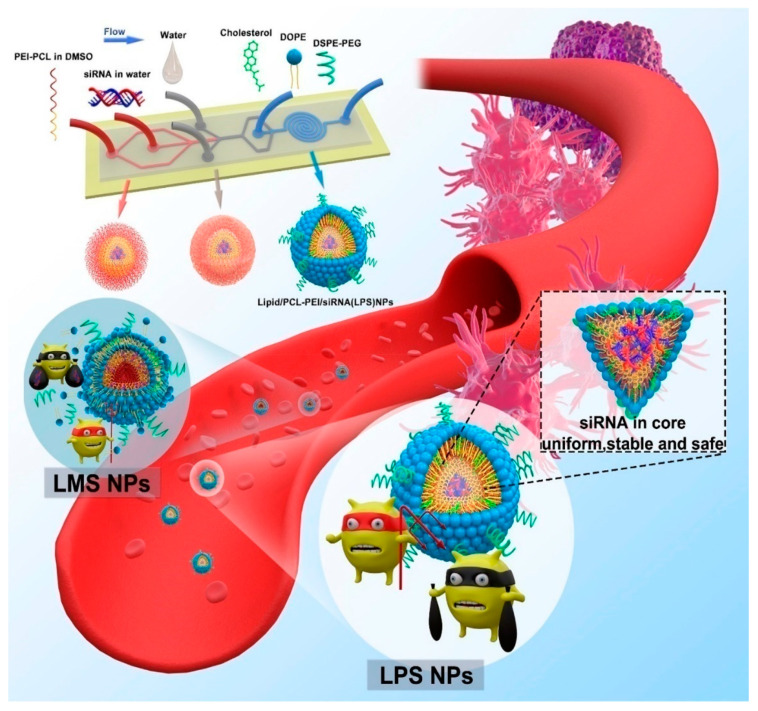
Synthesis of CSLPHNPs for delivery of siRNA with enhanced in vivo stability. The hybrid system synthesized by three-staged microfluidic technology with polyethyleneimine-polycaprolactone (PEI-PCL) in organic phase further coated over gene-loaded neutral lipid layer. Due to the lipid membrane coating, the core-shell contains reversed micelle via hydrophobic interaction, resulting in enhanced stability in vivo. Reprinted with permission from [136], Copyright © 2020, American Chemical Society.

**Figure 12 molecules-25-04377-f012:**
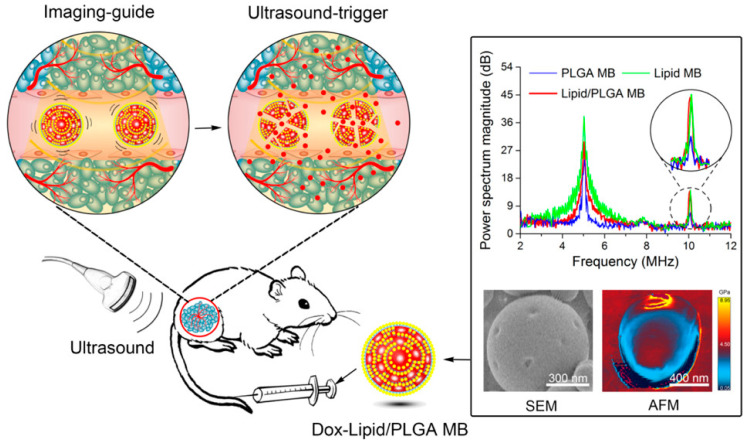
A multiparous lipid-PLGA hybrid nanoparticle synthesis for diagnosis and treatment. Reprinted with permission from [143], Copyright © 2019, American Chemical Society.

**Figure 13 molecules-25-04377-f013:**
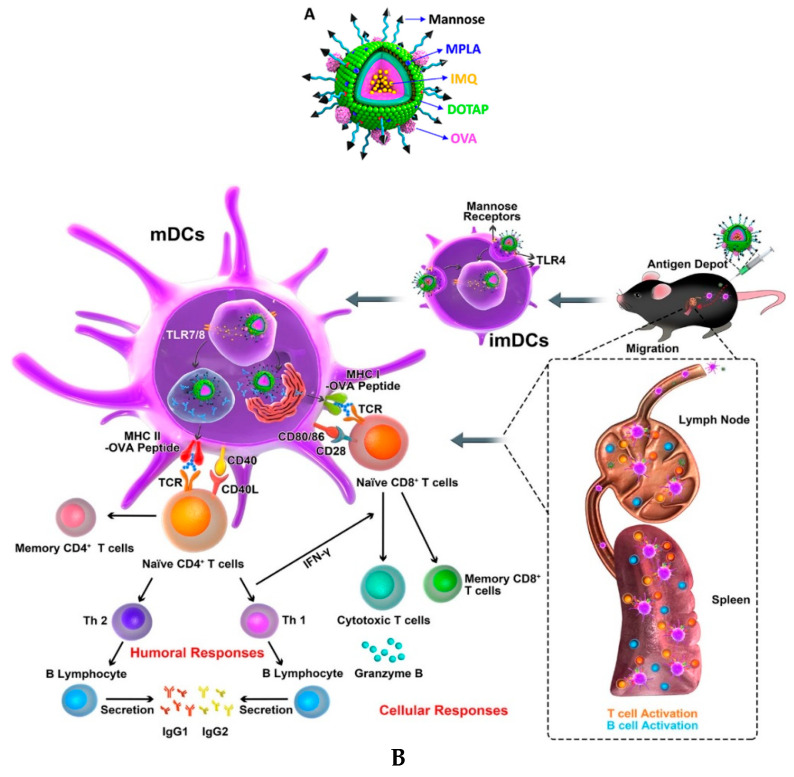
Multifunctional co-delivery by PLHNPs system in cancer immunotherapy. (**A**) Mannose receptor targeted hybrid vaccine system via PCL-PEG-PCL polymer with the help of cationic lipid DOTAP. The loaded IMQ (a TLR7/8 agonists), monophosphoryl lipid A (MPLA—a TLR4 agonist), and anionic OVA antigen containing the hybrid system were internalized by immature DC (dendritic cells). (**B**) Due to the strong antigen presentation by DC, MHC I and MHC II molecules increase the antigen peptide synthesis, along with the release of TLR4 and TLR7/8 in extracellular and intracellular response following endosomal escape. The upregulation of several inflammatory cytokines leads to innate immune system activation through highly adaptive Th1 cell immunity. The synergistic effect of both TLR agonist and the antigen promoted the activation of both B and T cells in both cellular and humoral immune responses. Reproduced with the permission from [147], Copyright © 2019, American Chemical Society.

**Table 2 molecules-25-04377-t002:** Examples of active targeted PLHNPs designed for various anticancer treatments.

Formulation		Delivery System	Targeting Receptors	Therapeutic Cargo	Cancer Treatment	Activity	References
Polymer	Lipids with Ligands						
Chitosan	Chondroitin sulphate, 1,2-distearoyl-sn-glycero-3-phosphoethanolamine-*N*{amino(polyethylene glycol 2000)- Folic acid (DSPE-PEG-FA)	PCLHNPs	Folate	Sorafenib	Liver cancer	Enhanced apoptosis with distorted nucleus	[74]
PLGA	Lecithin, mPEG-s-s-C_16_, DSPE-PEG-FA	MLPHNPs	Folate	DOX (Doxorubicin)	Cervical cancer	Higher stability and cytotoxicity toward KB cells	[78]
PCL-PEG-PCL	Soybean, DSPE-PEG-FA	CSPLHNPs	Folate	Paclitaxel	Murine carcinoma	65.78% growth inhibition compared to non-targeted one.	[75]
PLGA	Lecithin, DSPE-PEG, DSPE-PEG-FA	CSPLHNPs	Folate	Indocyanine green, cisplatin	Breast cancer	Effective tumor reduction with photothermal therapy	[76]
Polyaniline	1,2-dipamitoyl-sn-glycero-3-phosphocholine (DPPC), DSPE-PEG-FA	PCLHNPs	Folate	Doxorubicin	Breast cancer	Enhanced therapeutics and diagnosis with polyaniline	[121]
PLGA	DLPC, DSPE-PEG, DSPE-PEG-FA	MLPHNPs	Folate	Docetaxel	Breast cancer	polymer and lipid mixed ratio enhanced the DOX efficacy	[29]
PLA	Soybean phosphatidylcholine (SPC),1,2-dipalmitoyl -sn-glycero-3-phosphoethanolamine (DPPE)/DSPE-PEG	CSPLHNPs	Folate	Mitomycin C	Cervical cancer	High stability with water-soluble drug carrying phospholipid complex	[30]
PLGA	Soybean lecithin, PEG-RGD	CSPLHNPs	αβ-integrin	Paclitaxel, Cisplatin	Lung cancer	Excellent tumor reduction from- 1486- 263 mm^3^	[111]
PLGA	DSPE-PEG, PEG-iRGD	PCLHNPs	αβ-integrin	DOX, Sorafenib	Hepatocellular cancer	Enhanced tumor efficacy in HCC cells	[110]
PLGA	Lecithin/DSPE-PEG-OMe/DSPE-PEG-RGD	PCLHNPs	αβ-integrin	Docetaxel	Glioblastoma	2.69–4.13-fold increased anti-proliferative activity of DOPX	[122]
PLGA	Soybean lecithin, DSPE-PEG-mal, iRGD	PCLHNPs	αβ-integrin	Isoliquiritigenin (ISL)	Breast cancer	Effective delivery of ISL	[112]
Chitosan	Egg phospholipids, DPPE-HA	PCLHNPs	CD44	Moxifloxacin hydrochloride	Breast cancer	Enhanced ocular bioavailability, prolonged precorneal retention	[123]
PCL	DSPE-PEG, Lecithin, HA	MLPHNPs	CD44	Gallic acid, DOX	Blood cancer	Synergistic effect of drugs showed 77.7% tumor inhibition	[89]
PLGA	DOTAP-HA	PCLNPs	CD44	OVA (ovalbumin)	*-*	More powerful immune response	[92]
PLGA	DSPE-PEG-anti-EGFR aptamer CL4	PCLNPs	EGFR	Salinomycin	Osteosarcoma	Sustained drug release over 120 hrs.	[124]
PLGA	lecithin, DSPE-PEG-mal- anti-EGFR Fab	CSPLHNPs	EGFR	ADR (adriamycin)	Hepatocellular carcinoma	Reduced side population of HCC cells	[125]
PLGA	DOPA, D-α-tocopherol polyethylene glycol 1000 succinate (TPGS), AMD3100	PCLNPs	CXCR4	Sorafenib	Hepatocellular cancer	Reduced tumor infiltrated macrophages	[126]
PLGA	TPGS, egg-PC (phosphatidylcholine), Tf	PCLNPs	Transferrin	7α-APTADD (Aromatase inhibitor)	Breast cancer	Optimized lipid and polymer concentration showed control release of 7α-APTADD	[85]
PLGA	DSPE-PEG-Tf, Lecithin	CSPLHNPs	Transferrin	DOX	Lung cancer	Effective inhibition of tumor spheroids	[84]

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
