# Peer review of "Utilization of Polymer-Lipid Hybrid Nanoparticles for Targeted Anti-Cancer Therapy"

_molecules, 2020, doi:10.3390/molecules25194377_

Round 1
Reviewer 1 Report
This review summarizes the main recent advances of polymer-lipid hybrid nanoparticles. It is a well-structured and valuable work with the need of some minor corrections.
Comments:
Page 1, line 45: ’The NPs are approximately 1-1000 nanometers (nm) in diameter’ should be corrected to ’The NPs are 1-1000 nanometers (nm) in diameter according to a definition generally accepted in the area of microencapsulation.
Page 5, line 135: formatting error
Page 6, line 165: formatting error of subsection 2.2 title
Page 9, line 282: ’Hui Yang et al.’ There is not a citation number in the sentence. ’Hui’ must be deleted.
Page 12, line 381: ’Compare to…’ must be corrected to ’Compared to…’
Page 14, line 425: ’ERM’ must be defined.
Page 16, line 511: ’Enter’ error
Page 17, line 547: ’TET’ must be defined
A lot of references cited in the text as ’…et al’ cannot be found in the reference list or are present with different first author’s surname in the text, e.g. page 7, line 21: ’Sang et al’; page 17, line 560: ’Guowen et al’; page 24, line 706: ’Hui et al’; page 25, line 724: ’Ling et al’ . Furthermore, by most of them the number of reference cannot be found in the text in the same sentence, where they are.
Page 26, line 754: ’ iron oxide oxides’ should be corrected.
Author Response
Reviewer # 1:
This review summarizes the main recent advances of polymer-lipid hybrid nanoparticles. It is a well-structured and valuable work with the need for some minor corrections.
- Response: We would like to thank the reviewer for the careful and thorough reading of this manuscript and for the thoughtful comments and constructive suggestions, which help to improve the quality of this manuscript. Following your comments, we revised the entire manuscript citations presented in the text. The corrections have been included in the text as follows.
Comments:
Page 1, line 45: ’The NPs are approximately 1-1000 nanometers (nm) in diameter’ should be corrected to ’The NPs are 1-1000 nanometers (nm) in diameter according to a definition generally accepted in the area of microencapsulation.
Corrections: Page-1, line 45 – “The NPs are 1-1000 nanometers (nm) in diameter in size.”
Page 5, line 135: formatting error
Corrections: Page 5, line 135: We have corrected the formatting error - line 138.
Page 6, line 165: formatting error of subsection 2.2 title
Corrections: Page 6, line 165: We have corrected the formatting error of subsection 2.2 title - line 168.
Corrections: Page-6, line 200 – added the reference as [41].
Page 9, line 282: ’Hui Yang et al.’ There is not a citation number in the sentence. ’Hui’ must be deleted.
Corrections: Page-9, line 267- corrected the author's surname as Tahir et al, line 285- Huang et al., line 287- added reference as [58].
Page 12, line 381: ’Compare to…’ must be corrected to ’Compared to…’
Corrections: Page-13, line 363- corrected the author's surname as Gu et al, line 396- Guo et al., line 392- corrected the grammatical error as “compared to”.
Page 14, line 425: ’ERM’ must be defined.
Corrections: Page-14, line 411- corrected the author's surname as Wang et al. line -444- Shao et al.
Corrections: Page-15, line 437 – we defined the protein ERM - (ezrin, radixin, moesin),
Page 16, line 511: ’Enter’ error
Corrections: We have corrected the error.
Page 17, line 547: ’TET’ must be defined
Page-18, line 560- corrected the author's surname as Zhang et al, line 563- Zhang et al, line 575- Wang et al., line 562- tetrandrine (TET) defined.
A lot of references cited in the text as ’…et al’ cannot be found in the reference list or are present with different first author’s surname in the text, e.g. page 7, line 21: ’Sang et al’; page 17, line 560: ’Guowen et al’; page 24, line 706: ’Hui et al’; page 25, line 724: ’Ling et al’. Furthermore, by most of them, the number of references cannot be found in the text in the same sentence, where they are.
Page 26, line 754:’ iron oxide oxides’ should be corrected.
Corrections: We have corrected the full manuscript with their citations and authors' names. Revised citations are as follows-
Page-7, line 212- corrected the author's surname as Lee et al.
Page-8, line 248- corrected the author's surname as Khan et al.
Page-17, line 520 - corrected the author's surname as Hu et al, line 532-Zhang et al., line 538-Yong et al.
Page-19, line 581- corrected the authors surname as Li et al, added reference as [112], line 582- [113].
Page-20, line 616- corrected the author's surname as Gui et al.
Page-21, line 623- corrected the author's surname as Oh et al, line 656- Deshayes et al.
Page-25, line 710- corrected the author's surname as Diaye et al,
Page-26, line 721- corrected the author's surname Huang et al., line 738- Gao et al.
Page-27, line 760- corrected the author's surname as Zhao et al,
Page-28 - line 772- corrected the author's surname as Wo et al, line 787 - Gordijo et al, line 769- iron oxides.
Page-29, line 798- corrected the author's surname as Liu et al, line 805- Zhang et al.
Reviewer 2 Report
This review manuscript describes the application of polymer-lipid nanoparticles for anti-cancer therapy. It is an extensive work that in systematic way describes the usage of hybrid nanoparticles in cancer therapy. There is many original and well-prepared figures and tables. In my opinion before the publication in Molecules the abbreviation list and the table of content should be added to make this quite long article more clear for the reader.
Author Response
Reviewer # 2:
This review manuscript describes the application of polymer-lipid nanoparticles for anti-cancer therapy. It is an extensive work that in systematic way describes the usage of hybrid nanoparticles in cancer therapy. There is many original and well-prepared figures and tables. In my opinion before the publication in Molecules the abbreviation list and the table of content should be added to make this quite long article clearer for the reader.
- Response: Thank you for the reviewer’s insightful comment. Following your comment, we added the abbreviation list and table of contents into the manuscript. The revised manuscript is now provided as follows.
Addition:1
Page32, line- 927
Abbreviations:
PCLHNPs Polymer core lipid-shell hybrid nanoparticles
MLPHNPs Monolithic lipid-polymer hybrid nanoparticles
CSLPLHNPs Core-shell type hollow lipid-polymer lipid hybrid nanoparticle
PCLNPs Polymer-caged liposome hybrid nanoparticles
PLGA Poly (lactic-co-glycolic acid)
DSPE-PEG 1,2-distearoyl-sn-glycero-3-phosphoethanolamine-N- {amino (polyethylene glycol)}
TPGS D-α-tocopherol polyethylene glycol 1000 succinate
DDAB Didecyldimethylammonium bromide.
PCL Polycaprolactam
PLA Polylactic acid
DOTAP 1,2, -dioleoyl-3-triethylammonium-propane
DOPA Dioleoyl phosphatidic acid
DPPE 1,2-dipalmitoyl-sn-glycero-3-phosphoethanolamine
SPC Soybean phosphatidylcholine
DLPC 1,2-dilauroyl-sn-glycero-3-phosphocholine
DPPC 1,2-dipalmitoyl-sn-glycero-3-phosphocholine
HA Hyaluronic acid
FA Folic acid
5-FU 5-fluorouracil
DTX Docetaxel
VRS Vorinostat
FTY-720 Fingolimod hydrochloride
SRF Sorafenib
7α-APTADD 7α-(4´amino) phenylthiol-1,4-androstadiene-3,17-dione
ERM ezrin, radixin, moesin protein
AML Acute myeloid leukaemia
anti-CEA Anti-carcinoembryonic antigen
SnMP Mesoporphyrin
sFVA Antibody fusion protein
MHC I Major histocompatibility complex I
MHC II Major histocompatibility complex II
TLR4 Toll-like receptor4
TLR7 Toll-like receptor7
CTLA4 Cytotoxic T-lymphocyte-associated protein 4
CTL Cytotoxic T-lymphocyte
Addition:2
Page 32, line 958
Table of Contents
- Introduction
- General introduction
- Lipid nanoparticles
- Merits and demerits of Lipid nanoparticles
- Intelligent Hybrid lipid nanoparticles
- Polymer-Lipid hybrid nanoparticle
Types of polymer-lipid hybrid nanoparticle
- Polymer core lipid shell hybrid nanoparticles
- Monolithic lipid-polymer hybrid nanoparticles
- Core-shell type hollow lipid-polymer lipid hybrid nanoparticles
- Polymer caged liposomes
- Erythrocyte membrane-camouflaged polymeric hybrid nanoparticles
- Targeting
- Passive targeting
- Active targeting
- Active Targeting with Surface Engineered PLHNPs
- Receptors mediated targeting:
- Folate Receptor (FRs)
- Transferrin Receptor (TfRs)
- Cluster-of-Differentiation 44 Receptor (CD 44)
- Epidermal Growth Factor Receptor (EGFR)
- Biological ligands decorated PLHNPs
- Antibody
- Peptides
- Aptamers
- Dual-Targeting ligand
- Small molecules
- Applications of Polymer-Lipid Hybrid Nanoparticle
- Drug Delivery
- Gene therapy
- Imaging
- Immunotherapy
- Clinical Studies
- Summary, future perspective and challenges
Reviewer 3 Report
The manuscript reviews in an insightful way the interesting anticancer applications of lipid-polymeric hybrid nanoparticles. The review is generally well done and can be of interest to readers. There are not particular criticisms and therefore it is recommended publication and minor revision. Note some unformatted parts in the manuscript.
The following points need to be addressed though:
1) Title- What do the AA mean with the term intelligent? I suggest to change the title as this term is usually employed for a special kind of colloidal systems such as stimuli responsive gels or particles. The present review encompass a wider range of formulations and in particular targeted NP. Therefore, in order not to be misleading I suggest to delete the term intelligent. which is unnecessary.
2) The AA may discuss to a deeper level the technology transfer and scale up limits of such formulations that can halt their future development an commercialization. This issue is highly relevant as also demonstrated by the limited number of products reaching clinical trials and the market.
Author Response
Reviewer # 3:
The manuscript reviews in an insightful way the interesting anticancer applications of lipid-polymeric hybrid nanoparticles. The review is generally well done and can be of interest to readers. There are not particular criticisms and therefore it is recommended publication and minor revision. Note some unformatted parts in the manuscript.
The following points need to be addressed though:
1) Title- What do the AA mean with the term intelligent? I suggest to change the title as this term is usually employed for a special kind of colloidal systems such as stimuli responsive gels or particles. The present review encompass a wider range of formulations and in particular targeted NP. Therefore, in order not to be misleading I suggest to delete the term intelligent. which is unnecessary.
- Response: We thanks the reviewer’s great suggestion. We agreed with reviewer’s comments. We deleted the term “Intelligent” from the manuscript. The new title of the manuscript is given below-
“Utilization of Polymer-Lipid Hybrid Nanoparticles for Targeted Anti-cancer Therapy”
2) The AA may discuss to a deeper level the technology transfer and scale up limits of such formulations that can halt their future development and commercialization. This issue is highly relevant as also demonstrated by the limited number of products reaching clinical trials and the market.
- Response: Thank you for the reviewer’s insightful comment. Following your comment, we discussed a few key aspects in the perspective section with highlighting large scale-up point of view. The revised manuscript is now provided as follows.
Addition:
Page 31, line 892: However, the translation of PLHNPs into clinical applications is still in its infancy stage. The stability, toxicity, safety, and pharmacokinetics properties are the key aspects that should be focused in addition to the regulatory strategies necessary for predicting the potential risk and assessments of PLHNPs in the market. For example, there is a need for proper selection of solvent, material, and procedure before NPs fabrication. Specifically, the solvent used during synthesis can affect the stability of the NPs. During scaling-up, and they may induce toxic problems for the environment. In addition, during scaling up of PLHNPs, stability, binding ability, and circulating property could be compromised. To address these, there is a need to focus on developing solvent-free and easy scaling-up preparation methods suitable for the fabrication of PLHNPs. One of the major obstacles for the commercialization of PLHNPs is the need for expensive clinical studies, which could be overcome by investments from pharmaceutical and medical device corporations. Overall, the optimization of therapeutic requires, easy production steps, cost, scale-up and the addressing the regulatory concern can assist PLHNPs to attain the potential clinical applications.